# Methylation risk score of C-reactive protein associates sleep health with related health outcomes
Ziqing Wang [1,2] ✉, Danielle A. Wallace[1,2,3,4], Brian W. Spitzer[1], Tianyi Huang[5], Kent D. Taylor[6], Jerome I. Rotter [6], Stephen S. Rich [7], Peter Y. Liu [8], Martha L. Daviglus[9], Lifang Hou[9], Alberto R. Ramos[10], Sonya Kaur[10], J. Peter Durda[11], Hector M. González[12], Myriam Fornage [13,14], Susan Redline [2,3], Carmen R. Isasi [15] & Tamar Sofer[1,2,3,16] ✉

C-reactive protein (CRP) reflects inflammation status and is linked to poor sleep, metabolic and cardiovascular health. Methylation (MRS) and polygenic risk scores (PRS) reflect long-term systemic inflammation, and genetically-determined CRP, respectively. To refine understanding of inflammation-linked sleep and health outcomes, we construct PRS-CRPs using GWAS summary statistics and a previously-developed MRS-CRP in the Hispanic Community Health Study/Study of Latinos. Via survey-weighted linear regression, we estimate associations between blood-, PRS-, and MRS-CRP, with multiple sleep and health outcomes ($n = 2217$). MRS-CRP and PRS-CRPs are associated with increasing blood-CRP level by 43% and 23% per standard deviation. MRS-CRP is associated with obstructive sleep apnea (OSA) traits, long sleep duration, diabetes and hypertension, while PRS-CRPs were not. Blood-CRP level is associated with sleep duration and diabetes. Adjusting for MRS-CRP weakens OSA-diabetes/hypertension associations. Consequently, MRS-CRP is a stronger marker than blood-CRP and PRS-CRP to systemic inflammation associated with poor sleep and related comorbidities.

Sleep disorders are linked with impaired neurological function, metabolic and cardiovascular diseases (CVD)[1]. For instance, obstructive sleep apnea (OSA), characterized by episodic upper airflow narrowing or collapse[2], causes intermittent hypoxia, systemic inflammation and endothelial dysfunction. These physiological disturbances are considered key pathways linking OSA to increased risk for hypertension, cardiovascular and cerebrovascular diseases[3,4]. Sleep fragmentation and restriction-characteristics of many sleep disorders, including insomnia, have also been shown to elevate inflammation level and cardiovascular morbidity[5–7]. Furthermore, excessive daytime sleepiness (EDS) is a symptom of insufficient sleep or heightened sleep propensity and is also found to be associated with inflammation[8].

C-reactive protein (CRP), as a blood marker of systemic inflammation, is commonly used as a risk indicator for cardiometabolic diseases, including coronary heart disease, diabetes, and hypertension[9–11]. However, circulating CRP levels fluctuate following acute diseases, diminishing its stability and accuracy in estimating chronic inflammation. While sleep disorders, including OSA, have been associated with systemic inflammation[12], early studies reported conflicting results on associations of OSA with blood CRP

levels, likely due to limitations such as small sample size, study designs overlooking confounding factors including body mass index (BMI), detection threshold of CRP and so on[13,14].

DNA methylation (DNAm) is an epigenetic modification essential for transcriptional regulation and various fundamental biological processes[15], impacted by both genetics and environment[16]. While DNAm has site-specific effects, overall DNAm variability over time in adults is relatively stable[17]. Methylation risk scores (MRS) have been developed as weighted sums of DNAm at CpG sites and tested for their associations with health outcomes or biomarkers (such as aging or CRP) to infer exposures and disease states, similar to polygenic risk scores (PRS). Recently, an MRS for CRP (MRS-CRP) has emerged as a marker of interest for chronic inflammation, exhibiting enhanced test-retest reliability and stronger associations with long term health outcomes than circulating CRP[18–21]. That said, MRS-CRP is expected to capture chronic inflammation better than blood CRP, whose fluctuations are highly susceptible to circadian misalignment, acute medical condition and genetic predisposition[22,23].

CRP levels are also partly determined by genetics, with genetic loci associated with CRP explain approximately 10% of variation in blood CRP

level[21,24]. A previous study reported association of PRS-CRP with OSA and excessive daytime sleepiness (EDS), suggesting a potential causal relationship between chronic inflammation and symptomatic OSA with EDS[8]. Meanwhile further research and validation is needed to fill the knowledge gap regarding the role of genetics in the relationships between inflammation and sleep phenotypes.

The current study aims to investigate the relationships between circulating, epigenetic and genetic CRP patterns and common sleep problems (OSA; insomnia and EDS; sleep duration); as well as sleep-related comorbidities (hypertension, diabetes and cognitive decline); in the Hispanic Community Health Study/Study of Latinos (HCHS/SOL) cohort. We hypothesize that MRS-CRP will exhibit stronger associations, compared to blood CRP, with sleep measures capturing long-term conditions, such as OSA. We also hypothesized, based on the published study[8] that PRS-CRP will be associated OSA and with EDS. Additionally, we estimated the associations of sleep phenotypes with sleep-associated comorbidities, with and without adjusting for MRS-CRP, as we hypothesize that MRS-CRP could potentially explain the associations, as a common physiological link. Finally, to provide potential biological explanations of the link between inflammation and sleep-related phenotypes, we applied lasso penalized regression to identify key CRP-associated CpG sites associated with sleep traits and their comorbidities. The contributions of this paper include: (1) evaluation of CRP omics measures in a Hispanic population; (2) addressing the knowledge gap regarding the associations between MRS-CRP and sleep traits; (3) exploring associations between CRP omics measures with sleep, metabolic syndrome and cognitive function in a Hispanic population; (4) identification of specific CpG sites as potential links of the sleep-CRP association.

## Methods

Figure 1 illustrates the study design. We first constructed various PRS-CRPs in an independent cohort, the Multi-Ethnic Study of Atherosclerosis (MESA) using multiple methods and summary statistics to select the best-performing PRS-CRP based on its association with circulating CRP in MESA (Supplementary Methods). Since our primary interest is modeling associations in the Hispanic Community Health Study/Study of Latinos (HCHS/SOL) cohort, best-performing PRS-CRP and previously-developed MRS-CRP were then computed in HCHS/SOL, followed by association tests with blood CRP levels. We then evaluated associations of these CRP omics-measures with sleep, cognitive and metabolic comorbidities via survey linear

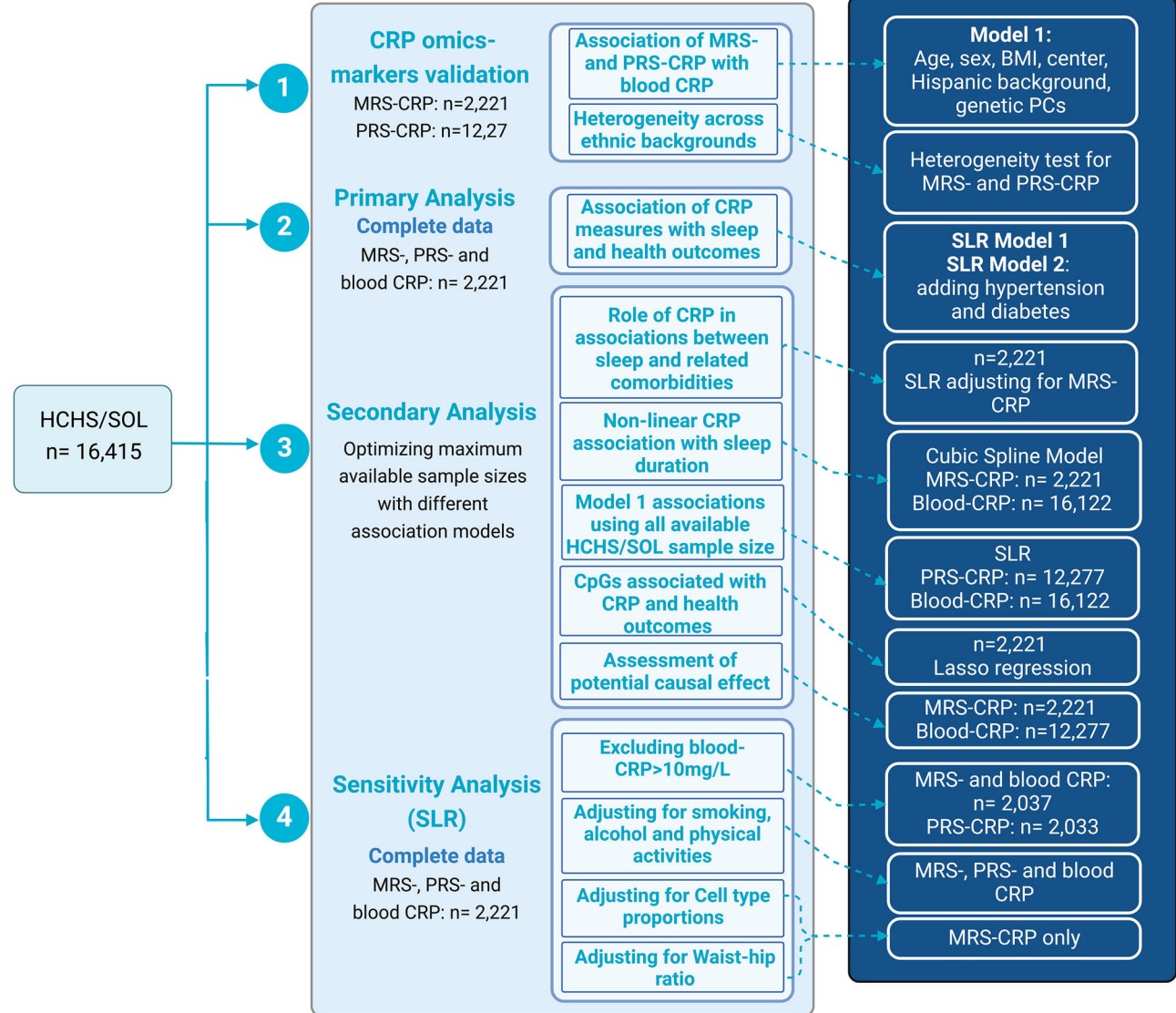

**Fig. 1 | Study design.** The associations of circulating CRP with Methylation risk score (MRS) and PRS constructed in HCHS/SOL was confirmed before evaluating the association of these CRP measures with sleep and other health outcomes. HCHS/ SOL: Hispanic Community Health Study/Study of Latinos. SLR: Survey linear regression. MRS: methylation risk score; PRS: polygenic risk score.

regression, and conducted secondary analyses to further explore the relationships between systemic inflammation and these health conditions from epigenetic and genetic perspectives.

## The HCHS/SOL dataset

The HCHS/SOL is a population-based cohort study with participants recruited from four metropolitan areas across the United States (Bronx NY, Chicago IL, Miami FL, and San Diego CA)[25]. The baseline HCHS/SOL exam took place during 2008-2011, and 16,415 participants were recruited. By design, participants represented diverse Hispanic/Latino origins (and genetic admixture) and had a wide age range (18–74 years old at baseline). Sociodemographic and lifestyle variables were self-reported, including age, sex, household income level, smoking status, cigarette and alcohol consumption, as well as self-identified Hispanic background. Obesity parameters BMI and waist-hip ratio were measured following standard procedure at baseline. BMI was calculated as body weight (kg) divided by height squared (m²), and waist-hip ratio as waist circumference divided by hip circumference.

In this work, the primary analysis focused on a subset of 2221 individuals from the Study of Latinos-Investigation of Neurocognitive Aging (SOL-INCA) ancillary study, who had methylation data available and further had sleep phenotypes, circulating CRP, and genetic data. Genetic data was available for 2217 of the primary analytic datasets. These participants also had their methylome profiled during Visit 2 (2014–2017). Some of the secondary analyses that did not use methylation data used larger subsets of the HCHS/SOL data. Sample size for the various primary and secondary analyses are provided in Fig. 1 and in reporting of results.

**Sleep assessment.** Multiple sleep phenotypes were assessed using self-reported questionnaires from the baseline visit[26], with no repeated measurements. Sleep duration was derived as weighted average of weekday and weekend sleep. Long and short sleep variables were derived from sleep duration as binary outcomes, with thresholds set at more than 9 h and less than 6 h. Scores indicative of insomnia and excessive daytime sleepiness were also measured using the 5-item Women's Health Initiative Insomnia Rating Scale (WHIIRS)[27] and the 8-item Epworth Sleepiness Scale (ESS)[28], respectively. OSA was assessed using a self-applied sleep apnea monitor (ARES Unicorder 5.2; B-Alert, Carlsbad, CA) that measured nasal airflow, position, snoring, actigraphy, and hemoglobin oxygen saturation (SpO2)[29] during overnight sleep. Obstructive respiratory events were defined as a 50% or greater reduction in airflow with associated desaturations of 3% or larger, lasting for at least 10 s. Respiratory event index (REI) was subsequently computed as the sum of such events per estimated sleep hour[26]. Given the importance of measures of hypoxia in characterizing OSA severity[30], we also studied several measures of overnight desaturation (minimum SpO2, average SpO2 and % Time SpO2 < 90 - the percentage of cumulative time with oxygen saturation below 90% in total sleep time).

**Assessment of C-reactive protein and relevant comorbidities.** In HCHS/SOL, high sensitivity CRP was assayed with a Roche Modular P Chemistry Analyzer (Roche Diagnostics Corporation, IN) using the immunoturbidimetric method (Roche Diagnostics Corporation, IN) during the baseline visit[31] (Visit 1). Demographic characteristics of the HCHS/SOL target population were summarized by clinical CRP risk groups for CVD ( < 1 mg/L [low risk], 1– < 3 mg/L [Borderline], and ≥ 3 mg/L [elevated risk])[32].

Diabetes status was identified (Visit 1) based on the American Diabetes Association definition, taking into account fasting serum glucose level (≥126 mg/dL), glucose tolerance test (≥200 mg/dL), medication use and hemoglobin A1c levels (HbA1c, ≥ 6.5%), whichever was available[33]. Hypertension was identified as the presence of either systolic (≥140 mmHg) or diastolic blood pressure (≥90 mmHg) exceeding the threshold, or by the use of antihypertensive medications[33]. Cognitive function score at baseline (Visit 1) was computed as the mean of the standardized scores of three

cognitive tests[34]: (1) Brief-Spanish English Verbal Learning Test (B-SEVLT; verbal episodic learning and memory); (2) Word Fluency test (WF; verbal fluency); (3) Digit Symbol Substitution test (DSS; processing speed and executive function)[34]. SOL-INCA ancillary study, conducted concurrently or within a year of Visit 2 (7 years after Visit 1, on average), repeated the same neuropsychological tests with additional two Trail Making Tests (TMT-A and B; executive function), which were also included in the calculation of the cognitive function score[34]. Cognitive change was defined as the difference between the cognitive function score at SOL-INCA and Visit 1.

**Genotype imputation and PRS construction.** HCHS/SOL participants were genotyped on a custom Illumina HumanOmni2.5-8v1-1array with 150,000 added custom variants, including ancestry informative single-nucleotide polymorphism (SNPs)[35]. Over 12,000 samples were imputed using the TOPMed imputation server[35]. As in MESA, MAF threshold was applied for variant exclusion in the SOL-INCA dataset, requiring MAF ≥ 0.01, where MAF was computed in HCHS/SOL. PRS-CRP for the HCHS/SOL dataset used in the study was computed using the same method as the best performing PRS-CRP selected in MESA, and PRS summation of PRS-CSx was computed with weights obtained from regression of CRP on ancestry-specific PRS-CSx PRSs using all MESA individuals and only MESA Hispanic individuals, as described in the Supplementary methods.

**Epigenome-wide methylation quantification and MRS construction.** Methylation profiling of whole blood samples from SOL-INCA participants across the genome was performed using the Infinium MethylationEPIC array (Illumina Inc, San Diego, CA). The quality control process involved filtering for sex and SNPs mismatches, as well as removal of control and blind duplicate samples. Subsequently, methylation data were normalized across all samples using the R package SeSAMe[36] to mask 105,454 probes, followed by dye bias correction, and normal-exponential deconvolution using out-of-band probes (Noob) background subtraction[37]. The final methylation values were calculated as the beta-values, defined as the percentage of methylation signal for each individual CpG site. The obtained beta-values underwent further correction for type-2 probe bias using beta mixture quantile normalization (BMIQ) from the WateRmelon package[38], followed by ComBat batch correction to account for position on array, slide and plate[39]. Cell type proportions were estimated using a reference-based method[40] based on the genome-wide methylation data.

Methylation risk scores (MRS) of CRP were constructed as weighted sums of each individual's DNAm values of pre-developed CpG sites[21] using MethParquet package (v0.1.0)[41] in R (v4.3.1)[42]. To validate that the most recently reported MRS-CRP (Hillary et al[21].) is also the most powerful MRS-CRP in Hispanic/Latino adults, we constructed multiple MRS-CRPs that were previously reported[18,21,43,44], and computed the correlation between blood CRP and the MRSs. For association analyses, the constructed MRS-CRP was standardized across the sample by subtracting the mean before dividing it by its standard deviation.

## Statistics and Reproducibility

All association analyses were performed using survey linear regression models (survey package v4.2.1)[45] in R (v4.3.1)[42] to account for the HCHS/SOL study design and generate inference applicable to the HCHS/SOL target population. Study design features addressed stratification, sampling weights, and non-response. Blood CRP level was log-transformed to account for the right skewed distribution (Figure S1). OSA related traits and cognitive phenotypes mentioned above were modeled as continuous variables, while excessive daytime sleepiness (EDS, ESS > 10), insomnia (WHIIRS ≥ 10), hypertension and diabetes were binary indicators.

**CRP omics-markers validation.** We first validated the associations of PRS- and MRS-CRP with blood CRP level using 12,636 and 2,221

observations with available data, respectively, by regressing blood CRP level over each omics marker while adjusting for age, sex, BMI, study center, Hispanic/Latino background, and first 5 genetic PCs (model 1). To assess whether the associations differ by Hispanic/Latino background, we performed regression analysis with no intercept and separately modeling the effect of the CRP measure in each Hispanic/Latino group (Central American, Cuban, Dominican, Mexican, Puerto Rican, South American and More than one/Other; i.e., as a separate term in the regression), and next applied Cochran's heterogeneity test while accounting for correlations between the effect estimates[46].

**Primary analysis.** We estimated the associations of sleep traits and related comorbidities, treated as outcomes, with each CRP measure, treated as exposures, (model 1) using 2,221 observations with available methylation and sleep measures. Model 2 was fitted further adjusting for hypertension and diabetes status. Statistical significance was based on multiple comparison adjustment with the Benjamini-Hochberg false discovery rate (FDR) correction on model 1 (FDR q-value < 0.05), grouped by each CRP variables. The FDR correction was employed to account for multiple testing, as the correlations among CRP measures and among sleep measures lead to non-independent tests, which could result in overly conservative Family-Wise Error Rate adjustment.

### Secondary analyses

**Association between sleep and related comorbidities.** For sleep phenotypes statistically associated with MRS-CRP in model 1, we assessed whether MRS-CRP accounts for some of their asociation with co-morbidities. To do this, we estimated and compared the sleep phenotype associations with diabetes, hypertension and cognitive scores with and without adjusting for MRS-CRP using the primary analytic dataset (2221 individuals). To assess whether the MRS-CRP associations with outcomes is driven in part by genetics, we compared MRS-CRP associations with health outcomes from model 1 with and without adjusting for PRS-CRP (n = 2217).

**Non-linear association with sleep duration.** To account for potential U-shaped association of sleep duration with MRS- and blood CRP, we fitted penalized cubic spline models using mgcv package (v1.9.1)[47] in R (v4.3.1)[42], adjusting for the same covariates as in model 1 using the primary analytic sample (2221 individuals). The basis dimension for the spline term (k) was selected by minimizing the Akaike Information Criterion (AIC). Same cubic spline model was fitted also using 16,122 samples with available circulating CRP levels to maximize statistical power.

**Analyses optimizing the available sample size.** While focusing on the methylation datasets allows for comparison between CRP measures, it limits inference for non-methylation measures by reducing power. Thus, we repeated the primary analysis (model 1) for blood (n = 16,122) and PRS-CRP (n = 12,277). Motivated by Huang et al.[8], we also estimated the associations of blood and PRS-CRP with OSA with EDS. We classified OSA cases with two thresholds (REI > 5, and REI > 15), as OSA compared to no OSA and moderate to severe OSA compared to mild and no OSA, respectively. According to the absence or presence of EDS cases, we tested the association of OSA cases with and without EDS with PRSs using all available 12,277 participants with appropriate data for the analysis[8].

**CpGs associated with CRP and health outcomes.** We applied lasso penalized regression on CRP-associated CpG sites constructing the MRS-CRP in association with sleep and health outcomes. The analysis was adjusted for age, sex, BMI, study center, first 5 genetic PCs and six cell-type proportions in blood estimated using DNA methylation data. We focused on phenotypes that were associated with MRS-CRP in primary analysis, including: REI, minimum and mean SpO2, insomnia, long sleep duration, diabetes and hypertension. We defined OSA with REI > 15, i.e.

moderate to severe OSA compared to mild and no OSA. Using this dichotomized outcome was more powerful in past lasso-based analysis of metabolomics and OSA, compared to using REI as the outcome[48].

**Assessment of potential causal effect.** One-sample Mendelian Randomization (MR) was performed using R package ivreg (v0.6.4)[49] in R (v4.3.1)[42] to assess whether blood CRP (n = 12,277) and MRS-CRP (n = 2221) show causal associations with sleep phenotypes, while leveraging the random allocation of genetic variants using all available samples. We used the few considered PRS-CRPs as instrumental variables. To assess whether there is effect of diabetes and REI on increasing MRS-CRP, we performed association analyses, modeling each of these measures separately as exposures, and the difference in MRS-CRP from Baseline to Visit 2 as the outcome, adjusting for model 1 covariates and time difference, in years, between the two visits.

### Sensitivity analyses

To account for acute inflammation, we conducted sensitivity analyses removing participants with blood CRP level above 10 mg/L[50] (n = 2,037 remaining). Given that DNAm profile derived from whole blood represents an average across all present cell types, and the association of systemic inflammation with central obesity, smoking, alcohol and physical activities, we repeated association analyses for MRS-CRP adjusting for (1) cell-type proportions; (2) waist-hip ratio; (3) smoking and alcohol consumption status, as well as total time per day spent engaging in physical activities (measured in metabolic equivalent of task (MET)-minutes).

### Reporting summary

Further information on research design is available in the Nature Portfolio Reporting Summary linked to this article.

## Results

### Demographics and outcome characteristics of study samples

Sample characteristics for MESA are summarized in Table S1. The number of participants in the three CVD risk groups is evenly distributed. There are race/ethnic differences in CRP levels, with the proportion of Chinese individuals decreasing by 18% while the proportion of Black and Hispanic individuals increased by 10% between the low risk and elevated risk groups.

Descriptive statistics of demographic characteristics based on the primary analytical HCHS/SOL sample are provided in Table 1. Overall, women constitute 65.8% of this sample, with higher female percentage, 75.7%, among participants with blood CRP level larger than 3 mg/L. Note that the high proportion of women is due to prioritization of individuals with Mild Cognitive Impairment (MCI) into the SOL-INCA methylation study, combined with higher MCI prevalence in women compared to men. Most OSA related measurements, along with hypertension and diabetes, exhibit greater severity and incidence rate in parallel with higher CRP blood level, whereas sleep duration, insomnia, daytime sleepiness and cognitive scores show little difference across the three risk groups (Table 1).

### Performance-based selection of PRS for CRP

Table S2 provides the number of SNPs used in different PRSs. The best performing PRS-CRP was selected as the one with the highest CRP blood level variance explained in MESA (Table S3). European ancestry-specific PRS-CSx explained more than 9% of the CRP level. Consequently, the PRS-CSx of European ancestry (PRS-EUR) was selected for subsequent association analysis in HCHS/SOL dataset. However, previous multi-cohort studies have shown that a weighted sum trained over multiple PRSs yields stronger association with related phenotypes[51]. Also, PRS weighted summation is the intended, developed use of PRS-CSx PRSs. Therefore, in HCHS/SOL, we also constructed a weighted sum of the five ancestry-specific PRS-CSx with adaptive weights (PRS-wsum), obtained using all MESA participants (PRS-wsum-All) and only MESA Hispanic participants (summation weights are provided in Table S4). Because PRS-wsum-all and PRS-wsum-Hispanic had comparable associations with blood CRP level in

**Table 1 | Demographic characteristics of SOL-INCA target population stratified by cardiovascular risk groups based on C-reactive protein levels (mg/L)**

| | Overall (n = 2221) | Low risk (<1) (n = 456) | Borderline (1-3) (n = 849) | Elevated risk (>3) (n = 916) |
|---|---|---|---|---|
| Age (Mean (SD)) | 56.66 (7.56) | 56.77 (7.81) | 56.57 (7.52) | 56.69 (7.49) |
| BMI (Mean (SD)) | 30.23 (5.63) | 26.86 (3.85) | 29.38 (4.79) | 32.70 (5.99) |
| Sex (%) - Female | 1435 (64.6) | 227 (49.8) | 520 (61.2) | 688 (75.1) |
| CRP (mg/L) (Mean (SD)) | 4.32 (9.13) | 0.60 (0.21) | 1.87 (0.57) | 8.44 (13.13) |
| **Background (%)** | | | | |
| Central American | 214 (9.6) | 44 (9.6) | 78 (9.2) | 92 (10.0) |
| Cuban | 380 (17.1) | 70 (15.4) | 126 (14.8) | 184 (20.1) |
| Dominican | 241 (10.9) | 49 (10.7) | 93 (11.0) | 99 (10.8) |
| Mexican | 754 (33.9) | 178 (39.0) | 313 (36.9) | 263 (28.7) |
| Puerto Rican | 41 (1.8) | 6 (1.3) | 19 (2.2) | 16 (1.7) |
| South American | 432 (19.5) | 77 (16.9) | 156 (18.4) | 199 (21.7) |
| **Sleep traits (mean (SD))** | | | | |
| REI | 8.16 (12.30) | 6.11 (9.48) | 7.41 (11.07) | 9.94 (14.32) |
| Min SpO2 | 85.75 (6.55) | 86.67 (6.04) | 86.26 (6.17) | 84.78 (7.03) |
| Average SpO2 | 96.29 (0.93) | 96.46 (0.77) | 96.34 (0.88) | 96.15 (1.03) |
| % Time SpO2 < 90 | 1.03 (3.11) | 0.69 (1.96) | 0.88 (3.01) | 1.36 (3.62) |
| ESS | 6.07 (5.06) | 5.95 (4.96) | 5.94 (4.90) | 6.25 (5.24) |
| WHIIRS | 7.96 (5.51) | 7.52 (5.53) | 7.78 (5.34) | 8.35 (5.63) |
| Sleep Duration (Mean (SD)) | 7.80 (1.42) | 7.70 (1.33) | 7.79 (1.33) | 7.85 (1.53) |
| **Comorbidities (%)** | | | | |
| Diabetes | 678 (30.5) | 101 (22.1) | 235 (27.7) | 342 (37.3) |
| Hypertension | 988 (44.5) | 163 (35.7) | 367 (43.2) | 458 (50.0) |
| **Cognitive traits (Mean (SD))** | | | | |
| Global Cognitive Baseline | 0.06 (0.74) | 0.06 (0.71) | 0.07 (0.76) | 0.06 (0.74) |
| Global Cognitive Change | −0.18 (0.54) | −0.14 (0.53) | −0.19 (0.54) | −0.18 (0.55) |

Means and SDs are weighted by SOL-INCA sampling weights to provide estimates that apply to the SOL-INCA study populations. ESS: Epworth Sleepiness Scale; REI: respiratory event index; SpO2: hemoglobin oxygen saturation; % Time SpO2 < 90: the percentage of cumulative time with SpO2 below 90% in total sleep time; WHIIRS: Women's Health Initiative Insomnia Rating.

HCHS/SOL (Figure S2), we focus PRS-wsum-all in subsequent association analyses. Table S5 provides the number of SNPs used for PRS-CSx construction in HCHS/SOL.

## CRP omics-markers validation

Both PRS and MRS-CRP are associated with blood CRP level (Fig. 2). Correlation analysis confirmed that the most recent MRS-CRP developed by Hillary et al. showed higher correlation with blood CRP level than others among HCHS/SOL participants (Figure S3). Of the 1468 CpGs previously identified for constructing this MRS[21], 1041 were available and used in MRS calculation. MRS-CRP showed an almost two-fold higher association (effect size 0.36) than the two PRSs (effect size 0.2) with log-transformed blood

CRP level. This means that each standard deviation (SD) increase in MRS-CRP corresponds to 43% percent increase blood CRP, while, one SD increase in PRS-wsum amounts to 23% percent increase in blood CRP level. Cochran's heterogeneity test suggests homogeneity of both MRS-CRP (p-value = 0.33) and PRS-CRP (p-value = 0.8) across Hispanic/Latino backgrounds in association with blood-CRP (see Table S6 for group-specific effect estimates).

## Primary analysis

Figure 3 summarizes the associations between the CRP measures and phenotypic outcomes with corresponding 95% confidence intervals. Interestingly, MRS-CRP is the only metric, modeled as exposure, to show statistically significant associations with REI (1.11 [0.36, 1.86]), minimum (−0.56 [−1.05, −0.08]) and average (−0.10 [−0.17, −0.04]) SpO2 (%) (Fig. 3A), implying that MRS-CRP correlates positively with OSA severity. Although not exhibiting significant associations with OSA traits, the coefficient of blood CRP level displays the same direction as that of MRS-CRP.

As shown in Fig. 3B, MRS-CRP is also associated with higher risks of insomnia (odds ratio (OR) = 1.19 [1.02, 1.38]) and long sleep duration (OR = 1.29 [1.08, 1.55]), while blood CRP level is associated with lower risks of EDS (OR = 0.81 [0.68, 0.96]), longer overall sleep duration (0.13 [0.04, 0.21]), and increased risk of long sleep duration (OR = 1.31, [1.13, 1.53]). Meanwhile, short sleep duration is not significantly associated with any of the CRP measures (Fig. 3B). On the other hand, PRS-CRPs are not associated with any sleep traits (Fig. 3).

Regarding other comorbidities, the association of greater genetic susceptibility to CRP, modeled via higher standardized weighted sum of PRS-CRP, and lower cognitive function at baseline (−0.05 [−0.1, −0.01]) fails to pass the multiple testing correction (Fig. 3A). MRS-CRP is the only CRP measure to have strong associations (q-value < 0.001) with both diabetes (OR = 2.44, [2.06, 2.90]) and hypertension (OR = 1.29, [1.10, 1.50]) (Fig. 3B).

The estimated associations of MRS-CRP with health outcomes decreased slightly in model 2 (Figure S4). More specifically, its associations with OSA and diabetes remain statistically significant after adjusting for cardiometabolic diseases, while its associations with minimum SpO2, insomnia, long sleep duration and hypertension became statistically insignificant (Figure S4).

## Secondary analyses

**Association between sleep and related comorbidities.** REI and minimum SpO2 were associated with diabetes, cognitive function and hypertension, and average SpO2 was associated with the two cardiometabolic diseases (Figure S5A). Notably, cognitive function score at baseline was positively associated with OSA severity, yet inversely associated with insomnia and long sleep duration (Figure S5A). Adjusting for MRS-CRP reduced the effect size of the associations with the two cardiometabolic disorders (Figure S5B), suggesting that MRS-CRP captures the inflammatory component shared by sleep phenotypes and these comorbidities. Interestingly, the strongest change was observed for diabetes, whereas adjusting for MRS-CRP had a much smaller impact on associations with hypertension and cognitive scores (Figure S5). Adjusting for PRS-CRP essentially had no effect on the associations of MRS-CRP with OSA phenotypes, long sleep and metabolic comorbidities (Figure S6).

**Non-linear association with sleep duration.** Our cubic spine model results showed a J-shaped association between sleep duration and CRP markers where optimal sleep time (around 7 h) corresponds to lowest level of both MRS- and blood CRP level (Figure S7). As sleep duration extends, the increase in levels of CRP markers become more pronounced (Figure S7). The protective association of short sleep on blood CRP persisted in analyses that utilized data from the larger sample, while the rate of increase in blood CRP level associated with longer sleep becomes less pronounced (Figure S7B).

**Fig. 2 | Association of polygenic risk score (PRS) and methylation risk score (MRS) for C-reactive protein (CRP) with blood CRP level.** PRS-EUR: PRS-CRP developed as the European-specific PRS from the PRS-CSx analysis; PRS-wsum: PRS-CRP taken as the weighted sum of ancestry-specific PRS-CSx PRSs.

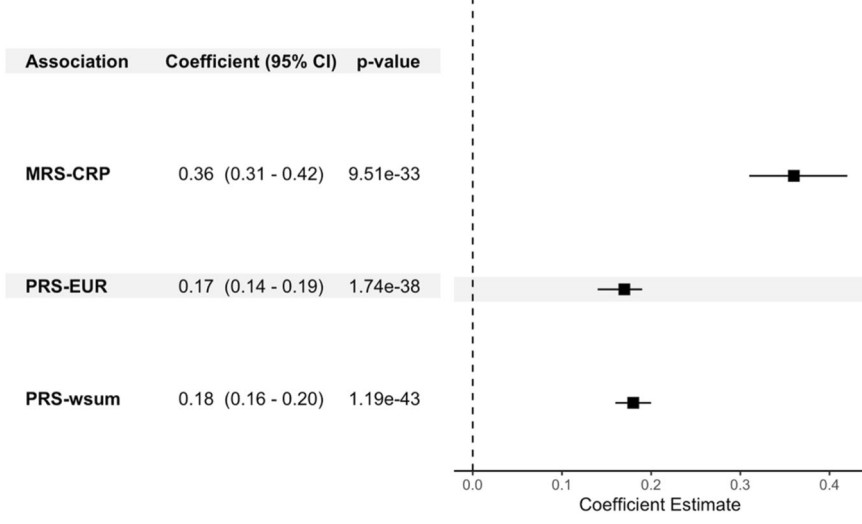

## Model 1 associations using all available HCHS/SOL sample size.

Associations of blood CRP level with REI (0.41 [0.16, 0.66]) and cognitive function at baseline (−0.04 [−0.06, −0.01]) become statistically significant after increasing the sample size to all HCHS/SOL participants, yet its association with EDS become statistically insignificant (Figure S8). When it comes to PRS-CRPs, their association $p$-values remain unchanged across these health outcomes (Figure S8). Table S7 provides the results of association of PRS-CRPs with OSA risk with and without EDS, in two comparison scenarios, where OSA was classified as OSA (REI > 5) and moderate to severe OSA (REI > 15). In contrast to the previous study[8], our computed PRS-CRP (PRS-EUR and PRS-wsum) were nominally associated with reduce likelihood of OSA with EDS, compared with no OSA (Table S7), however these associations are not statistically significant when accounting for multiple testing. Interestingly, PRS computed based on weights from the previous study[8] (PRS-ty) is significantly associated with higher risks of moderate to severe OSA with EDS (OR = 1.26 [1.01, 1.56]) and moderate to severe OSA (OR = 1.10 [1.01, 1.21]) (Table S7).

## CpGs associated with both CRP and health outcomes.

Out of the list of the 1,041 CpG sites used in the construction of the MRS-CRP score, lasso penalized regression identified 66 and 12 CpG sites as associated with OSA and minimum SpO2 (log transformed), respectively, yet 0 CpG sites for average SpO2, insomnia and long sleep duration (Table S8-S9). In terms of diabetes and hypertension, 55 and 6 CRP associated CpG sites were selected by lasso penalized regression (Table S10-S11). Among them, 6 CpG sites are shared between OSA-related measures, and 5 shared across OSA and hypertension and/or diabetes. Table 2 provides the CRP-related CpG sites that were selected by Lasso as associated with at least two of the studied phenotypes.

## Assessment of potential causal effect.

The F-statistics of the PRS-EUR and PRS-wsum for blood CRP were 340.7 and 373.9, respectively, suggesting that they are strong instrumental variables for blood CRP. However, neither PRS-CRP is a statistically strong instrument for predicting MRS-CRP (F-statistics <10), which may lead to unreliable causal effect estimates[52]. The one-sample MR results indicated that blood CRP has no statistically significant causal effect on the CRP-associated sleep phenotypes (Table S12). Neither diabetes (−0.12 [−0.24, 0.003]) nor REI (0.002 [−0.002, 0.006]) was found to be associated with changes in MRS-CRP between baseline and Visit 2.

## Sensitivity analysis.

After excluding 184 participants with blood CRP level above 10 mg/L, the effect sizes of the associations between CRP measures with health outcomes decreased, particularly for blood CRP level (Figure S9). Nevertheless, associations between MRS-CRP and REI (0.91 [0.22, 1.61]), mean SpO2 (−0.08 [−0.14, −0.01]), long sleep duration (OR = 1.30 [1.06, 1.06]), diabetes (OR = 2.46 [2.02, 3.00]) and hypertension (OR = 1.28 [1.09, 1.52]) remain statistically significant, with minimal changes observed for metabolic comorbidities (Figure S9).

The effect sizes of MRS- and PRS-CRP with health outcomes remained largely unchanged after adjusting for smoking status, alcohol consumption and physical activity levels, whereas blood-CRP level became statistically significant with REI (0.09 [0.05, 0.14]) (Figure S10). Moreover, the association between circulating CRP and risk of diabetes and long sleep duration diminished to be statistically non-significant (Figure S10).

Adjusting for cell-type proportions and waist-hip ratio yielded comparable results in association significance level to that of primary analysis for MRS-CRP (Figure S11). Surprisingly, adjusting for waist-hip ratio instead of BMI increased considerably the effect size of the association between MRS-CRP and most studied outcomes except for insomnia, long sleep duration and diabetes (Figure S11).

## Discussion

This study demonstrates that omics measures of CRP perform well in association with CRP in a population of Hispanic/Latino adults, which were not examined before for these associations. Despite the heterogeneity (both genetic and lifestyle-related) of this population, there was no evidence of heterogeneity of biomarker association with CRP across self-reported Hispanic/Latino background groups. We validated the stronger associations of MRS-CRP with long-term sleep and metabolic conditions, such as OSA and diabetes, compared to blood CRP, which showed modest associations. Adjusting for MRS-CRP attenuated the associations between sleep traits and metabolic conditions, suggesting systemic inflammation as either a mediating or a shared causal factor. However, in contrast to MRS-CRP, PRS-CRP exhibited minimal associations with sleep and related health outcomes, suggesting limited role in influencing these inflammation-related phenotypes. Further, one-sample Mendelian randomization analysis did not support a causal association of CRP on neither sleep, diabetes, nor hypertension. Overall, this work demonstrates strong association of inflammation, modeled via MRS-CRP, with sleep-related health outcomes. Secondary analysis selecting specific CpG methylation sites in association with sleep-related and metabolic outcomes also advances the understanding of potential epigenetic mechanisms underlying the relationship between these health outcomes and inflammation.

MRS-CRP stands out as the only CRP measure associated with insomnia and OSA severity markers including REI, minimum and mean SpO2. Blood CRP level, on the other hand, is associated with REI after

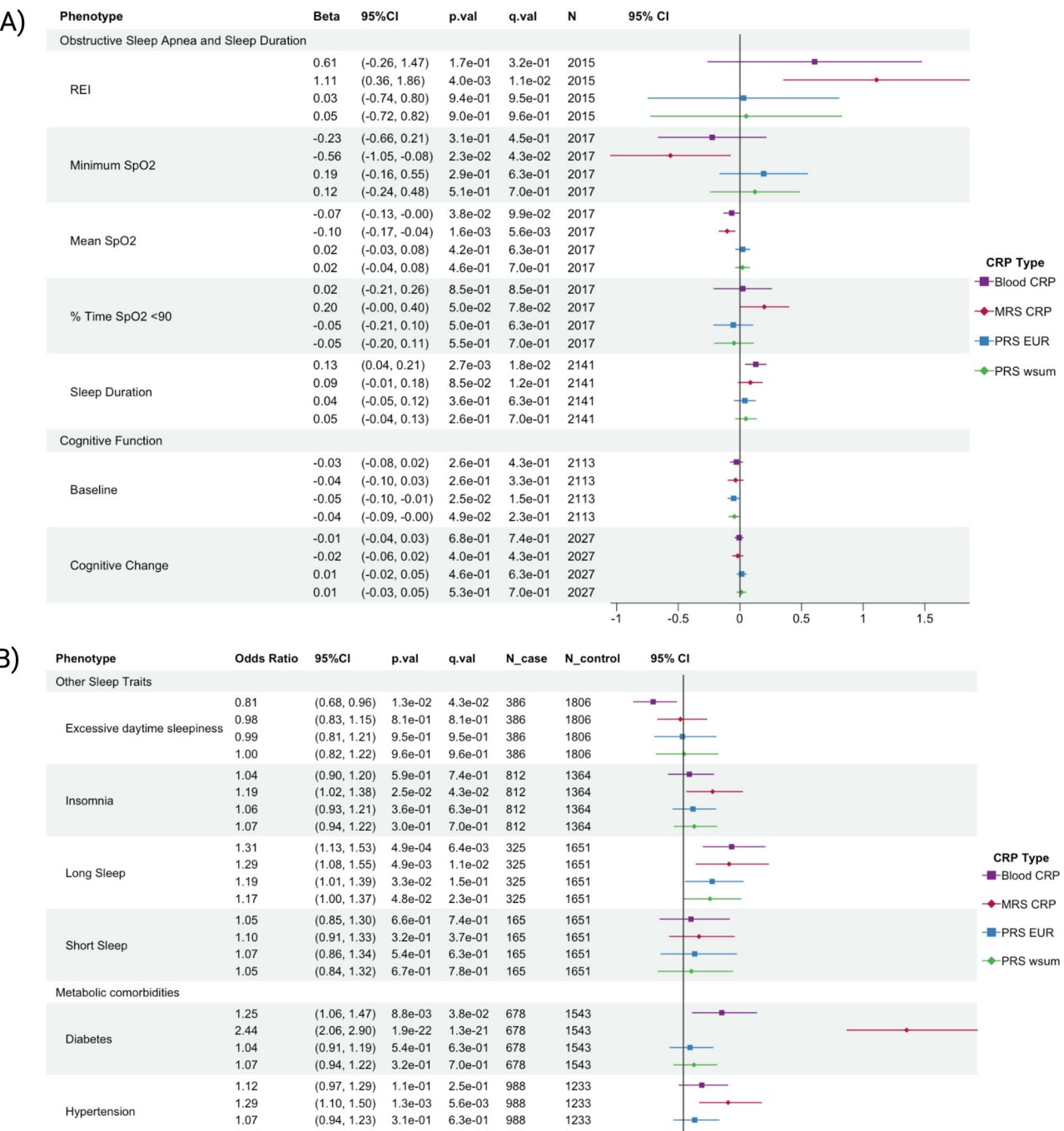

**Fig. 3 | Forest plot showing associations between C-reactive protein (CRP) measures and measured health outcomes. A** Obstructive sleep apnea (OSA), sleep duration and cognitive traits; (**B**) odds ratio for binary cardio-metabolic and other sleep outcomes. From left to right model coefficients or odds ratio in case of binary outcomes, 95% confidence interval (95% CI), *p* value (p.val) and FDR corrected q value (q.val). MRS: methylation risk score; PRS: polygenic risk score; REI: respiratory event index; SpO2: hemoglobin oxygen saturation; % Time SpO2 < 90: the percentage of cumulative time with SpO2 below 90% in total sleep time.

increasing the sample pool to all HCHS/SOL participants, or once adjusting for physical activity levels, smoking and alcohol consumption status. A previous study reported no association between average SpO2 and blood CRP in participants with OSA[53]. Our findings in the general Hispanic population further support that average SpO2 is likely a less specific marker for OSA compared to MRS-CRP. Notably, since the CpG sites for MRS-CRP construction were developed using non-Hispanic participants[21], our results supported not only the superior performance of MRS-CRP to circulating CRP in measuring chronic inflammation and related comorbidities[18–21], but also its transferability across different racial groups.

Several factors could impact the performance of circulating CRP. In addition to its short-term fluctuations in response to environmental and physiological changes, CRP molecules exist in two structural forms, pentameric and monomeric[54]. The former is synthesized in the liver and regulated by proinflammatory cytokines. While the latter is dissociated from the pentameric CRP at local inflammation sites such as atherosclerotic plaque, further escalating inflammatory activities[55]. However, current high sensitivity assays measure only the pentameric form[54], thus limiting the representation of overall inflammatory status. In contrast, MRS-CRP, derived from a mixture of cellular populations in the blood, reflects

**Table 2 | CpG sites selected by Lasso regression overlapped between at least two phenotypes studied**

| CpG | Chromosome | Gene Name | Phenotypes |
|------|-----------|-----------|------------|
| cg00572560 | 10 | | OSA, Diabetes |
| cg00574958 | 11 | CPT1A | OSA, Min SpO2, Diabetes |
| cg03246954 | 19 | MKNK2 | OSA, Min SpO2 |
| cg06690548 | 4 | SLC7A11 | OSA, Hypertension |
| cg14476101 | 1 | PHGDH | OSA, Min SpO2, Hypertension |
| cg14656297 | 9 | FXN | OSA, Min SpO2 |
| cg19693031 | 1 | TXNIP | OSA, Diabetes, Hypertension |
| cg23281327 | 10 | | OSA, Min SpO2 |
| cg23440058 | 3 | KALRN | OSA, Min SpO2 |

*OSA* obstructive sleep apnea, *SpO2* hemoglobin oxygen saturation.

epigenetic changes resulting from cellular responses to inflammation. This makes MRS-CRP a promising marker for stratifying population risk across numerous outcomes, including chronic inflammation, sleep and metabolic health, thereby enhancing accuracy and clinical inference. That said, it is beneficial to verify these associations between MRS-CRP and health outcomes using incident data for future research.

CRP has emerged as a useful clinical marker for OSA and cardiovascular comorbidities, due to its independent association with apnea severity and hypertension[56]. Given that OSA is recognized as an antecedent risk factor for secondary hypertension[57], the modest change in association between OSA traits and hypertension after adjusting for MRS-CRP suggests a relatively small contribution of inflammation to the elevated blood pressure. Nevertheless, the association between sleep outcomes and diabetes were greatly attenuated and even no longer statistically significant after adjusting for MRS-CRP, which is in alignment with the mitigating effect of CRP over the association of OSA severity and incident diabetes[58]. These findings indicate that systemic inflammation may function as a mediator or a common factor underlying the relationship between sleep phenotypes and diabetes.

Both MRS- and blood CRP markers manifest a J-shaped association with sleep duration, supporting the previously reported association between sleep duration and CRP markers[6,59]. Previous research has also linked diabetes, hypertension, depression and low-grade inflammation[60] to prolonged sleep time, which is generally considered as an indicator of poor health condition[61]. Congruent with our findings, the relationship between short sleep duration and CRP levels is no longer statistically significant after adjusting for confounding variables[6,59], probably due to the strong associations of sex, race/ethnicity, and age with both extreme sleep durations and blood CRP level[6]. Interestingly, the slightly protective effect of short sleep on CRP levels in Hispanic (Figure S7) and Asian population[6] warrants future studies to replicate these results using larger sample sizes among different races/ethnic groups, and to investigate how inflammation interplays with sleep duration and cardiovascular comorbidities.

Our results validated that the PRS-CRP associates with blood CRP in a Hispanic/Latino population. On the other hand, the non-significant associations (accounting for multiple testing) observed for PRS-CRP with sleep and health outcomes indicate a weak relationship between genetic component of CRP with related comorbidities. Given that the PRS-CRP explained over 9% of the variance of circulating CRP levels, it is more likely that CRP is a consequence of these health conditions[9,62], rather than the other way around. That being said, there was a nominally significant association (OR = 1.26, p-value = 0.04) of the PRS-CRP constructed using a limited number of genome-wide significant SNPs, mimicking the one used by Huang et al.[8], with moderate-to-severe OSA with EDS, compared with no OSA. Further research on the link between OSA and inflammation is needed to continue refining these results.

Another limitation of current study stems from the potential bias in self-reported sleep measurements, which often fall short of capturing

context-specific sleep patterns influenced by environmental and lifestyle factors. Moreover, our one-sample MR and the association analysis for change in MRS-CRP are limited by a relatively small sample size (n = 2221), which makes it challenging to detect true relationship between PRS-CRP and MRS-CRP, as well as the true effect between disease onset and long-term change in MRS-CRP. Overall, it remains to be determined whether the inflammation-associated DNAm causes or results from sleep and metabolic disorders, or if a third scenario exists where both are affected by a shared underlying causal factor. In light of the established associations, the underlying epigenetic mechanisms behind the observed inflammatory processes and how they relate to sleep and its co-morbidities still need to be elucidated.

To address this question, we applied lasso regression to select key CRP-associated CpGs related to health outcomes. Consequently, 6 CpGs were found to be shared between OSA traits and metabolic disorders. Various studies confirmed the strong association of cg19693031 and cg00574958 with incident and prevalent type 2 diabetes[63–67]. Specifically, hypomethylation of cg19693031 and cg00574958 enhances thioredoxin-interacting protein (*TXNIP*) expression, which activates monocytes towards an inflammatory state[65,68]. The negative coefficient of these loci supports its protective role against the disease onset (Table S8-S11), suggesting a possible connection between OSA and diabetes through hyperglycemia-induced inflammation[69] and energy homeostasis[66].

Solute carrier family 7 member 11 (*SLC7A11*)-cg06690548 and phosphoglycerate dehydrogenase (*PHGDH*)-cg14476101 have been frequently reported together as showing protective association against blood pressure[70,71]. Such protective effect was also observed in HCHS/SOL, with hypermethylation at both sites associated with improved REI, SpO2 levels, and lower hypertension risk (Table S8-S9, Table S11), through modulation of metabolic reprogramming[70,72] and lipid homeostasis[73].

Overall, these lasso-selected CpG sites reiterate the important role of DNAm in the pathogenesis underlying complex diseases, and may serve as potential biomarkers for inflammatory stress associated with metabolic syndrome and OSA traits, bridged by fat, glucose and energy homeostasis. The large effect sizes of some CpGs for REI and minimum SpO2, such as *MKNK2*-cg03246954, *PFDN2*-cg00816397, *ATF2*-cg02622866, and *ARSA*-cg07453718, indicate strong associations with OSA, and potential applicability as disease biomarkers (Table S8-S9). Future research could benefit from confirming causality of these associations as well as their directionality, and expanding to populations with other ancestries to test efficacy of the epigenetic predictor of CRP.

## Conclusion

A CRP measure based on DNA methylation status at select CpG sites shows stronger relationship with circulating CRP level than PRS-CRP. Moreover, MRS-CRP outperforms both genetic and circulating CRP in association with several OSA-related sleep traits and metabolic comorbidities, while its associations with other sleep and cognitive measures were mixed. Our results also suggest a possible shared mechanism between sleep phenotypes and metabolic syndrome, related to systemic inflammation and modeled by MRS-CRP, particularly diabetes. Consequently, the incorporation of MRS-CRP, versus circulating CRP, in clinical assessment has the potential to improve population risk stratification for adverse health outcomes.

## Data availability

HCHS/SOL data are available through application to the data base of genotypes and phenotypes (dbGaP) accession phs000810, or via data use agreement with the HCHS/SOL Data Coordinating Center (DCC) at the University of North Carolina at Chapel Hill, see collaborators website: https://sites.cscc.unc.edu/hchs/. TOPMed-MESA freeze 10 WGS data are available by application to dbGaP according to the study specific accession: MESA: "phs001211". MESA phenotype data are available from dbGaP according to the MESA accession: MESA: "phs000209". Summary statistics from GWAS of CRP were downloaded from the PAN-UKBB project (https://pan.ukbb.broadinstitute.org/; phenocode 30710) and BioBank

Japan PheWeb at https://pheweb.jp/pheno/CRP. PRS-CRP alleles and weights will be provided at the Github repository: https://github.com/ZWangTen/CRP_markers_association.

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

## Acknowledgements

We thank the staff and participants of MESA and HCHS/SOL for their important contributions. HCHS/SOL investigators website - http://www.cscc.unc.edu/hchs/. The work was supported by National Heart Lung and Blood Institute (NHLBI) grants R01HL161012 to T.S., National Institute on Aging (NIA) grants R01AG080598 to T.S., RF1AG061022 to M.F. and H.M.G., and R01AG048642, R56AG048642 and R01AG075758 to H.M.G. The Hispanic Community Health Study/Study of Latinos is a collaborative study supported by contracts from the National Heart, Lung, and Blood Institute (NHLBI) to the University of North Carolina (HHSN268201300001I/N01-HC-65233), University of Miami (HHSN268201300004I/N01-HC-65234), Albert Einstein College of Medicine (HHSN268201300002I/N01-HC-65235), University of Illinois at Chicago (HHSN268201300003I/N01-HC-65236 Northwestern Univ), and San Diego State University (HHSN268201300005I/N01-HC-65237). The following Institutes/Centers/Offices have contributed to the HCHS/SOL through a transfer of funds to the NHLBI: National Institute on Minority Health and Health Disparities, National Institute on Deafness and Other Communication Disorders, National Institute of Dental and Craniofacial Research, National Institute of Diabetes and Digestive and Kidney Diseases, National Institute of Neurological Disorders and Stroke, NIH Institution-Office of Dietary Supplements. MESA and the MESA SHARe project are conducted and supported by the National Heart, Lung, and Blood Institute (NHLBI) in collaboration with MESA investigators. Support for MESA is provided by contracts HHSN268201500003I, N01-HC-95159, N01-HC-95160, N01-HC-95161, N01-HC-95162, N01-HC-95163, N01-HC-95164, N01-HC-95165, N01-HC-95166, N01-HC-95167, N01-HC-95168, N01-HC-95169, UL1-TR-000040, UL1-TR-001079, UL1-TR-001420. MESA Family is conducted and supported by the National Heart, Lung, and Blood Institute (NHLBI) in collaboration with MESA investigators.

Support is provided by grants and contracts R01HL071051, R01HL071205, R01HL071250, R01HL071251, R01HL071258, R01HL071259, and by the National Center for Research Resources, Grant UL1RR033176. The provision of genotyping data was supported in part by the National Center for Advancing Translational Sciences, CTSI grant UL1TR001881, and the National Institute of Diabetes and Digestive and Kidney Disease Diabetes Research Center (DRC) grant DK063491 to the Southern California Diabetes Endocrinology Research Center. Molecular data for the Trans-Omics in Precision Medicine (TOPMed) program was supported by the National Heart, Lung and Blood Institute (NHLBI). Genome sequencing for "NHLBI TOPMed: Multi-Ethnic Study of Atherosclerosis" (phs001416.v2.p1) was performed at Broad Institute Genomics Platform (HHSN268201500014C, 3U54HG003067-13S1). Core support including centralized genomic read mapping and genotype calling, along with variant quality metrics and filtering were provided by the TOPMed Informatics Research Center (3R01HL-117626-02S1; contract HHSN268201800002I). Core support including phenotype harmonization, data management, sample-identity QC, and general program coordination were provided by the TOPMed Data Coordinating Center (R01HL- 120393; U01HL-120393; contract HHSN268201800001I). We gratefully acknowledge the studies and participants who provided biological samples and data for TOPMed.

## Author contributions

Z.W. performed the data analysis and wrote the manuscript. T.S. conceived of the study, procured funding support, provided advisory guidance, revised and edited the manuscript. B.W.S. and T.H. developed CRP PRSs. K.D.T., J.I.R., S.S.R., and J.P.D. acquired and curated MESA data. M.L.D., A.R.R., S.K., S.R., C.R.I. were involved in HCHS/SOL study design, data collection and curation. H.M.G., M.F. designed and executed the SOL-INCA and SOL-INCA methylation studies. Z.W., D.A.W., B.W.S., T.H., K.D.T., J.I.R., S.S.R., J.P.D., P.Y.L., M.L.D., A.R.R., S.K., S.R., L.H., C.R.I., H.M.G., M.F., T.S. critically reviewed and approved the manuscript.

## Competing interests

The author declare no competing of interests.

## Ethics statement

H.C.H.S./S.O.L.: This study was approved by the institutional review boards (IRBs) at each field center, where all participants gave written informed consent, and by the Non-Biomedical IRB at the University of North Carolina at Chapel Hill, to the HCHS/SOL Data Coordinating Center. All IRBs approving the study are: Non-Biomedical IRB at the University of North Carolina at Chapel Hill. Chapel Hill, NC; Einstein IRB at the Albert Einstein College of Medicine of Yeshiva University. Bronx, NY; IRB at Office for the Protection of Research Subjects (OPRS), University of Illinois at Chicago.

Chicago, IL; Human Subject Research Office, University of Miami. Miami, FL; Institutional Review Board of San Diego State University. San Diego, CA. MESA: All MESA participants provided written informed consent, and the study was approved by the Institutional Review Boards at The Lundquist Institute (formerly Los Angeles BioMedical Research Institute) at Harbor-UCLA Medical Center, University of Washington, Wake Forest School of Medicine, Northwestern University, University of Minnesota, Columbia University, and Johns Hopkins University. This work was approved by the Beth Israel Deaconess Medical Center Committee on Clinical Investigators (protocol #2023P000538).

## Additional information

[1]Cardiovascular Institute, Beth Israel Deaconess Medical Center, Boston, MA, USA. [2]Department of Medicine, Harvard Medical School, Boston, MA, USA. [3]Division of Sleep Medicine and Circadian Disorders, Department of Medicine, Brigham and Women's Hospital, Boston, MA, USA. [4]Gangarosa Department of Environmental Health, Rollins School of Public Health, Emory University, Atlanta, GA, USA. [5]Laboratory of Epidemiology and Population Sciences, Intramural Research Program, National Institute on Aging, Baltimore, MD, USA. [6]The Institute for Translational Genomics and Population Sciences, Department of Pediatrics, The Lundquist Institute for Biomedical Innovation at Harbor-UCLA Medical Center, Torrance, CA, USA. [7]Department of Genome Sciences, University of Virginia School of Medicine, Charlottesville, VA, USA. [8]Division of Genetics, Lundquist Institute at Harbor-UCLA Medical Center, Torrance, CA, USA. [9]Department of Preventive Medicine, Northwestern University Feinberg School of Medicine, Chicago, IL, USA. [10]Department of Neurology, University of Miami Miller School of Medicine, Miami, FL, USA. [11]Department of Pathology and Laboratory Medicine, University of Vermont, Burlington, Vermont, USA. [12]Department of Neurosciences and Shiley-Marcos Alzheimer's Disease Center, University of California, San Diego, La Jolla, CA, USA. [13]Human Genetics Center, Department of Epidemiology, Human Genetics, and Environmental Sciences, School of Public Health, The University of Texas Health Science Center at Houston, Houston, TX, USA. [14]Brown Foundation Institute of Molecular Medicine, McGovern Medical School, University of Texas Health Science Center at Houston, Houston, TX, USA. [15]Department of Epidemiology & Population Health, Department of Pediatrics, Albert Einstein College of Medicine, Bronx, NY, USA. [16]Department of Biostatistics, Harvard T.H Chan School of Public Health, Boston, MA, USA. ✉e-mail: zwang14@bidmc.harvard.edu; tsofer@bidmc.harvard.edu

