## [Transparent Peer Review file · Communications Biology]

Methylation risk score of C-reactive protein associates sleep health with related health outcomes

Corresponding Author: Dr Ziqing Wang

Version 0:

Reviewer comments:

Reviewer #1

(Remarks to the Author)

The present manuscript examines the association of various sleep parameters with CRP levels and the polygenic risk scores and methylation risk scores for CRP as well as related comorbidities.

Various analyses are performed, not all of which necessarily contribute to answering the research question, but make the manuscript very long (e.g. 6 pages of methods) and difficult to follow.

On the other hand, critical aspects such as causality and direction of the observed associations deserve more attention.

Title

The authors should consider rewording the title. It is not clear what "C-reactive protein omics-measures" is referring to, especially as the methylation risk score is named separately

Abstract

- Introduction: The first sentence of the introduction is too strong. It indicates the DNA methylation is a better measurement as CRP itself, which is an oversimplification. Suggest introducing the differences between blood, genetic, and epigenetic CRP markers to let readers easily understand their roles. "a accurate means" should be "an accurate means".

- Methods: information on sample size, outcome and exposure definitions, models and confounders should be provided; "... established the association between ..." should be "associations".

- Results: information on the effect strength should be included, to evaluate whether the effect size is relevant

- Conclusions: The results do not support why MRS-CRP is proven as a promising estimate for systemic inflammation. In the introduction section, it is mentioned that MRS-PRS was an accurate means of assessing chronic inflammation, and the blood CRP was a marker of systemic inflammation.

Introduction

- The first sentence already implied causality by describing the sleep disorders as risk factors, but it would be possible to have a bi-directional association. This aspect requires more attention throughout the manuscript and should be addressed where possible and elaborated on in the introduction and discussion

- The statement "MRS for CRP (MRS-CRP) has emerged as a more accurate marker for chronic inflammation than circulating CRP" is misleading and should be rephrased. The authors probably refer to the inflammation score but it is not possible that the MRS is more accurate than the measurement itself

- The difference between chronic and acute CRP should be considered in the hypothesis and more information on the different CRP markers provided.

- Page 3, 1st paragraph, last sentence: "...found associated with inflammation" should be "...found to be associated with inflammation".

-

Methods & Results

- The first paragraph of the methods section seems to describe the flow depicted in Figure 1. Both is too generic to be helpful and should be revised to provide accurate details. The figure is also inconsistent with the order of the methods and results described in the text. Full names of the abbreviations in the footnote should be added.

- The methods section is extremely lengthy and difficult to follow. It should always be clear which model with which adjustment is used, which is the outcome and exposure and what is their distribution. Also the numbers must be clarified, as

there are major discrepancies between the numbers reported in the methods and results sections. Two previous papers are briefly referred to. Here it should be made clear what has been done and what the novel aspects are.

- hs CRP cut-off: To evaluate CRP as a pathway, it is important to focus on the sub-clinical values and exclude those with acute response. This would require removing all the participants with values above 3mg/L, ideally also above 1mg/L. Which threshold was applied to determine acute infections?
- PRS: It is not clear why such a large proportion and effort is going towards the PRS calculation, which does not seem to be necessary in such detail and could easily be reduced to a standard PRS approach. Potential reasons for the PRS results (or lack of association) should also be elaborated on in the discussion.
- hs CRP distribution: Usually in the general population, the hs CRP values are skewed. How was determined if the log-transformation leads to a normally distributed variable and the assumptions for linear regression are not violated? The authors should also check the results presentation. A simple back transformation as indicated by the description of Figure 2 is not possible. It is rather more appropriate to report the effect as percent change or similar to make interpretation easier.
- Adjusting for MRS: The rationale for this approach should be clarified. If the authors speculate that the CRP is the relevant mechanism (which is supported by the MRS but not PRS results) the approach to adjust for MRS might lead to collider bias, as adjustment on the causal chain might be problematic. Maybe other methods such as mediation analysis would be more appropriate.
- HCHS/SOL: The study design should be clarified. Are repeated measurements available? At which time-points (years) were the data collected?
- MRS calculation: There is a lot of effort in the PRS calculation. It is unclear why "only" one MRS was calculated and clumping and thresholding was not done here? The approaches used here should correspond for both scores. The participants from HCHS/SOL cohort study represented diverse Hispanic/Latino origins (and genetic admixture). However, the CpGs for constructing the MRS for CRP were derived from the reference 21 paper by Hillary et al., which did not include the individuals of Latino race/ethnicity. The potential impact of this should be addressed in the discussion. It is also not clear why the authors then test again individual CpGs and focus on these results in the discussion. Both methods have their strength and limitations and the analysis should be focussed on the approach which helps to address the research question.
- Page 6, 2nd paragraph: please add the full name of PCs for the first time.
- Page 6, 3rd paragraph: "cohort study with participants of participants" participants was duplicated.
- Sleep: why was a cutoff of 6 hours used to define short sleep, rather than the recommended 7 hours?
- Cell type proportions: this is very relevant. It should be stated in the methods how the cell type proportions were assessed.
- It is unclear which models were used. Is it always the "survey" approach? Why is this necessary? The weights are usually applied to obtain representative results, but this is not required here. Again, more details on the study design should be provided
- "associations of PRS- and MRS-CRP with blood CRP level were first validated": What validation has been done?
- Confounder selection: The confounders for models 1 and 2 are reported twice on page 8. It is not clear why only a very basic set of confounders was chosen and other relevant factors such as smoking, physical activity, alcohol intake are not considered. Also, in addition to the lack of a PRS effect, the strong impact of waist-to-hip ratio indicates substantial residual confounding.
- Role of ethnicity: It is highly appreciated that the present study is not limited to Caucasians only. As different PRS are calculated and tested, this aspect should also be acknowledged in the discussion
- Given the large number of analyses, the unit and effect estimate should always be given when reporting the results.
- Multiple testing: Why did the authors use FDR adjustment? How was this calculated with the large number of different models? This is usually applied for explorative analysis. In this case, a clear hypothesis is tested. The Bonferroni correction with the number of independent tests might be more appropriate.
- Page 9, last paragraph: "Overall, women constitute 65.8% of this sample, with higher female percentage, 75.7%, among participants with blood CRP level larger than 3 mg/L." The numbers in this sentence were not consistent with those in the Table 1. For example, women constitute 65.8% of this sample, but the Table 1 shows 1435 females, and the percentage should be $1435/2221 = 64.6\%$. This percentage 1435 (58.4%) in Table 1 was also not correct. "75.7%, among participants with blood CRP level larger than 3 mg/L" was also wrong. Many numbers and percentages in Table 1 do not match and should be double-checked.
- Page 13, Figure 3: The text on the right is not aligned with the forest plot on the left. Full names of the abbreviations should be added in the footnote.
- Page 14, 3rd and 4th paragraphs: several papers were cited to compare the findings with previous research. Since this is the Results section, it would be more appropriate to focus solely on describing the results rather than including discussions or comparisons with other studies.
- Figure S5: The figure shows a J shaped association, although this does not seem to be significantly different from a linear association. What is the estimated df? Was the non-linear term significant?

Discussion

- The discussion should start with an overall summary of the findings to address the hypothesis and include a strength and limitations section.
- The direction and strength of the effect and causality should be discussed.
- What is the novel aspect and clinical relevance? CRP is already included in the standard clinical routine, what could be improved here or would this provide a therapeutic target?
- "PFDN2-cg00816397, ATF2-cg02622866 and ARSA-cg07453718" were not found in the Table 2.
- The conclusion should be revised and focused on the results

Reference

References are in various formats, which should be harmonised.

Reviewer #2

(Remarks to the Author)

Peer-review for: Analysis of C-reactive protein omics-measures associates methylation risk score with sleep health and related health outcomes

This is an interesting study concept on how sleep apnea is related to chronic inflammation as measured by CRP. My main concerns are twofold. First, I'm not sure that the MRS for CRP was constructed adequately. Second, my interpretation of the plots is slightly different than those of the authors (I see mostly null results). I attempted to also one avenue to improve some of the expected biases, which I believe would be easy to perform by the authors.

Major points:

- 1) It is well explained that the CRP PRS was built in a separate cohort (MESA) than the one it was tested on (HCHS/SOLA). The authors actually take great care in ensuring that some their analyses do not overfit. However, my current understanding of the MRS is that it was computed on the HCHS/SOL cohort, the same one it was tested on. Is that so? This is important, as if it were the case then the association between MRS and the different traits could be falsely amplified.
- 2) When I look at Figure 3, I feel like the number of bars that do not cross the null is grossly expected for the number of analyses. Further, even if assuming that no multiple testing errors are shown, I have a hard time wrapping my head around some results. For example, if the CRP PRS is indeed associated with higher AHI, then how to reconcile this with a lower daytime sleepiness?
- 3) One way to potentially alleviate some of the problems above would be to use the CRP PRS as a statistical instrument in a one-sample MR. That is measure the PRS calculated in MESA in the HCSC/SOLA participants. Then use a two-stage linear regression model with this calculated PRS. Since the authors have primary data access, this should be easy to do using the appropriate statistical package (R has good two-stage linear regression packages, for instance). This is a rigorous way of doing one-sample Mendelian randomization, assuming that the PRS explains enough of the variance in CRP (i.e. has an F-value greater than 10, as a rule of thumb).

Reviewer #3

(Remarks to the Author)

This is a well-executed and technically sound study. The authors have applied appropriate and timely analytical approaches throughout. While they did not find an association between their genetic risk score and health-related outcomes, this result is intriguing in itself. Furthermore, their identification of novel associations between the methylation risk score and sleep-related outcomes is particularly noteworthy, as well as the expected associations with cardiometabolic outcomes. Overall, the study offers valuable insights and makes a meaningful contribution to the field.

As I understand it, there are three major CRP EWAS meta-analyses. It would be valuable to assess the stability of the MRS-CRP score based on 1,041 CpGs from Hillary et al. (2024) in comparison to the 218 CpGs identified by Ligthart et al. (2016) and the 1,511 CpGs reported by Wielscher et al. (2022). Specifically, it would be informative to examine how these scores, derived from earlier EWAS studies, are associated with both blood CRP levels and sleep traits within the HCHS/SOL study.

It would also be highly interesting to explore the differences in CRP associations between the PRS and MRS across different ethnicities. Stratifying the MRS ~ blood CRP associations by the detailed population background in your study could provide valuable insights and would be a compelling addition to the analysis.

For Figure 3, please add clarifications in the legend to define terms such as AHI and mean SpO₂, ensuring that the figure can be understood independently. Additionally, since the PRS is not significantly associated with any outcomes, I suggest removing the PRS weighted sum associations, as they do not seem to contribute meaningfully to the narrative. Lastly, the alignment of the forest plot with the table on the left side is off, making it difficult to read; this should be adjusted for better clarity.

For Figure 1, it would be helpful to clarify the source of the input CpGs for the MRS—specifically, indicating which paper they are derived from. Additionally, the figure would benefit from a clearer distinction between the processes involving the PRS and MRS, as well as the datasets used for each. This would enhance the figure's overall clarity and make it easier for readers to follow the methodology.

Since you have already conducted an association analysis between sleep outcomes and diabetes, adjusting for MRS, it would be valuable to also perform a mediation analysis. This could help determine whether the mediation effect of the MRS on the relationship between sleep outcomes and diabetes is significant.

Version 1:

Reviewer comments:

Reviewer #2

(Remarks to the Author)

The authors have addressed my concerns.

I also checked authors' responses to Reviewer comments. The large majority of comments from reviewer 1 were about narrative problems and were easy to address. I will not comment on these any further.

The only points where I think there was a more significant methodological issue were I believe addressed well by the authors. These were:

For the reviewer's comment about excluding participants with a CRP above 3, I actually agree with the authors that in clinical practice a cutoff about 10 is generally used as an indication for acute inflammation. Some labs use a cutoff of 5, but I think a sensitivity analysis of 10 (as the authors now performed) is appropriate (and also supported their results)

For the reviewer's comment about the need for normally distributed variables in linear regression, I also agree with the authors that this is a common misconception about linear regression. Only the error terms need to be normally distributed, and even then results are generally robust to deviation from this assumption (as explained by the authors in their responses). For the reviewer's comment on the choice of adjustment for multiple testing, I agree with the choice of FDR method that the authors used.

So all in all I am happy with publication at this point

Response to review of COMMSBIO-24-5677 “Methylation risk score of C-reactive protein associates sleep health with related health outcomes”

We thank the reviewers and the editor for reviewing our manuscript and providing many useful suggestions. We believe that the manuscript is substantially improved in the revisions. We would also like to note that we updated the use of the term “AHI” (apnea-hypopnea index) to “REI” (respiratory even index), which is the term now being used for home sleep apnea testing.

Responses to reviewer #1 begin on page **1**

Responses to reviewer #2 begin on page **17**

Responses to reviewer #3 begin on page **19**

Reviewer #1 (Remarks to the Author):

The present manuscript examines the association of various sleep parameters with CRP levels and the polygenic risk scores and methylation risk scores for CRP as well as related comorbidities.

Various analyses are performed, not all of which necessarily contribute to answering the research question, but make the manuscript very long (e.g. 6 pages of methods) and difficult to follow.

On the other hand, critical aspects such as causality and direction of the observed associations deserve more attention.

Response: thank you for your comprehensive review, we appreciate you taking the time to carefully assess all aspects of our manuscript.

Title

The authors should consider rewording the title. It is not clear what “C-reactive protein omics-measures” is referring to, especially as the methylation risk score is named separately

Response: Thank you for the suggestion! We have updated the title to be “Methylation risk score of C-reactive protein associates sleep health with related health outcomes”.

Abstract

- Introduction: The first sentence of the introduction is too strong. It indicates the DNA methylation is a better measurement as CRP itself, which is an oversimplification. Suggest introducing the differences between blood, genetic, and epigenetic CRP markers to let readers easily understand their roles. “a accurate means” should be “an accurate means”.

Response: Thank you for the suggestion. The statement about “accurate means” was removed. We also updated the introduction part to be:

“C-reactive protein (CRP) reflects inflammation status and is linked to poor sleep, metabolic and cardiovascular health. Methylation (MRS) and polygenic risk scores (PRS) reflect long-term systemic inflammation, and genetically-determined CRP, respectively.

- Methods: information on sample size, outcome and exposure definitions, models and confounders should be provided; “...established the association between ...” should be “associations”.

Response: Thank you very much for the suggestion. However, due to the 150 words limit of the journal, we have updated the methods section in the abstract as following:

“To refine understanding of inflammation-linked sleep and health outcomes, we constructed PRS-CRPs using GWAS summary statistics and a previously-developed MRS-CRP in the Hispanic Community Health Study/Study of Latinos. We applied survey-weighted linear regression to estimate associations between blood-, PRS-, and MRS-CRP, with multiple sleep and health outcomes (n= 2,217).”

- Results: information on the effect strength should be included, to evaluate whether the effect size is relevant

Response: We have added the effect size of the associations between methylation and polygenic risk score with circulating C-reactive protein:

“MRS-CRP and PRS-CRPs were associated with increasing blood-CRP level by 43% and 23% per standard deviation.”

- Conclusions: The results do not support why MRS-CRP is proven as a promising estimate for systemic inflammation. In the introduction section, it is mentioned that MRS-PRS was an accurate means of assessing chronic inflammation, and the blood CRP was a marker of systemic inflammation.

Response: Thank you very much for the suggestion. We’ve reframed our wording to:

“Consequently, MRS-CRP is a stronger marker than blood-CRP and PRS-CRP to systemic inflammation associated with poor sleep and related comorbidities”

We would like to note that we chose to still write that MRS-CRP is a stronger marker of systemic inflammation associated with poor sleep, compared to blood CRP (and PRS-CRP). Following your comment we do realize that our results do not entirely prove this statement. But we think they support it, because MRS-CRP is developed specifically as a marker of systemic inflammation, poor sleep (and especially OSA) has been shown a few times to be associated with systemic

inflammation, so that observing stronger association of MRS-CRP than blood CRP is consistent with it being a stronger marker of systemic inflammation.

Introduction

- The first sentence already implied causality by describing the sleep disorders as risk factors, but it would be possible to have a bi-directional association. This aspect requires more attention throughout the manuscript and should be addressed where possible and elaborated on in the introduction and discussion

Response: Thank you for this comment. This is correct – the relationship between sleep and other outcomes is complicated. While various poor sleep measures have been associated with incident outcomes (incident hypertension, diabetes, cognitive measures), the underlying relationship is indeed sometimes bidirectional, as well as due to common causes. To address your comments, we first changed the wording of the first sentence in the introduction to be weaker in the way it states the relationship between sleep and other outcomes: “Sleep disorders are linked with impaired neurological function, metabolic and cardiovascular diseases (CVD)”.

- The statement “MRS for CRP (MRS-CRP) has emerged as a more accurate marker for chronic inflammation than circulating CRP” is misleading and should be rephrased. The authors probably refer to the inflammation score but it is not possible that the MRS is more accurate than the measurement itself

Response: Thank you for the suggestion. To clarify, our goal was to distinguish chronic from acute effect. For example, blood CRP may better capture inflammation following a UTI, but perhaps MRS-CRP captures better the long-term impact of obesity, which is a long-term condition. We rephrased the sentence to be: “Recently, an MRS for CRP (MRS-CRP) has emerged as a novel marker for chronic inflammation, exhibiting enhanced test-retest reliability and stronger associations with long term health outcomes than circulating CRP^{18–21}. That said, MRS-CRP is expected to capture chronic inflammation better than blood CRP, whose fluctuations are highly susceptible to circadian misalignment, acute medical condition and genetic predisposition”.

- The difference between chronic and acute CRP should be considered in the hypothesis and more information on the different CRP markers provided.

Response: Thank you for the suggestion! First, we clarified the difference between the CRP markers in the abstract (“Methylation (MRS) and polygenic risk scores (PRS) reflect long-term systemic inflammation, and genetically-determined CRP, respectively.”). The introduction includes explanations about the MRS (e.g. explained that DNA methylation is relatively stable over time), and why it is expected to capture chronic inflammation better than blood CRP (Paragraph 3). We also added information about the PRS in the introduction:

“A previous study reported association of PRS-CRP with OSA and excessive daytime sleepiness (EDS), suggesting a potential causal relationship between chronic inflammation and symptomatic OSA with EDS⁸. Meanwhile further research and validation is needed to fill the knowledge gap regarding the role of genetics in the relationships between inflammation and sleep phenotypes.”

Finally, we reframed our hypothesis in the introduction in accordance with your suggestion, which we completely agree with. Accordingly, we’ve added specific hypothesis in the introduction:

“We hypothesize that MRS-CRP will exhibit stronger associations, compared to blood CRP, with sleep measures capturing long-term conditions, such as OSA. We also hypothesized, based on the published study⁸ that PRS-CRP will be associated OSA and with EDS. Additionally, we estimated the associations of sleep phenotypes with sleep-associated comorbidities, with and without adjusting for MRS-CRP, as we hypothesize that MRS-CRP could potentially explain the associations, as a common physiological link.”

- Page 3, 1st paragraph, last sentence: “...found associated with inflammation” should be “...found to be associated with inflammation”.

Response: Thank you! We’ve made the change accordingly.

Methods & Results

- The first paragraph of the methods section seems to describe the flow depicted in Figure 1. Both is too generic to be helpful and should be revised to provide accurate details. The figure is also inconsistent with the order of the methods and results described in the text. Full names of the abbreviations in the footnote should be added.

Response: Thank you for pointing this out. We considered your comment, which we generally agree with. We decided to keep the overview paragraph with less details compared to the rest of the methods, because we believe it helps explain the analysis, which has many components. However, your point is well taken and we re-created the figure, so that the figure provides accurate detail, and the organization of the text is changed for consistency with the figure. Note that we made a larger change to address a few of your other comments. Specifically, we moved some parts relating to the development of PRS-CRP, as they are less central to this work, to the supplement to simplify the presentation.

Also, we added abbreviations in the footnote Figure 1.

- The methods section is extremely lengthy and difficult to follow. It should always be clear which model with which adjustment is used, which is the outcome and exposure and what is their distribution. Also the numbers must be clarified, as there are major discrepancies between the numbers reported in the methods and results sections. Two previous papers are briefly referred to. Here it should be made clear what has been done and what the novel aspects are.

Response: Thank you for pointing these out. We did a few things to make the presentation clearer. (1) We have moved the MESA and PRS construction section to supplementary information. (2) We revised Figure 1 in a way that we believe is more clear and accurate. It now provides sample sizes for various analyses that uses different subsets of the data. We think that it is now made clear that the primary dataset is the intersection of individuals who have methylation, genetic, and sleep data, and that secondary analyses have different sample sizes that are maximized beyond the primary dataset based on specific measures used in each analysis. (3) We renamed “CRP association analysis” to “Statistical analysis” and reorganized it according to subsections in accordance with Figure 1.

- **hs CRP cut-off:** To evaluate CRP as a pathway, it is important to focus on the sub-clinical values and exclude those with acute response. This would require removing all the participants with values above 3mg/L, ideally also above 1mg/L. Which threshold was applied to determine acute infections?

Response: Thank you for the comment. According to literature review, we performed sensitivity analysis excluding participants with CRP level above 10mg/L as showing acute infection/). We also updated the manuscript with this analysis in the methods and results section.

In more detail, we first would like to point out that, as seen in Table 1, most of the HCHS/SOL population (80%) has hs CRP values 1 and higher, and 41% have CRP values >3. Clearly, excluding so many individuals will risk reduced power, and increased uncertainty in estimation and in conclusions that can be drawn. We reviewed the literature to understand which CRP values are observed in a general population (such as the HCHS/SOL). According to this literature review, CRP levels were found to be higher in African and Hispanic compared to White population (Nazmi and Victora, 2007; Albert et al. 2004), with median/mean CRP level around or higher than 3 mg/L (Lundin et al. 2024; Reiner, et al. 2012; Stowe et al. 2009; Vintimilla et al. 2019). While it is possible that CRP level < 1mg/L is ideal, given the patterns in our data as well as other datasets, we think that restricting analyses to individuals with such low values, and even restricting to <3mg/L, would limit the interpretation of the results tremendously.

Reference:

- Nazmi A, Victora CG. Socioeconomic and racial/ethnic differentials of C-reactive protein levels: a systematic review of population-based studies. *BMC Public Health*. 2007;7:212. Published 2007 Aug 17. doi:10.1186/1471-2458-7-212
- Albert MA, Glynn RJ, Buring J, Ridker PM. C-reactive protein levels among women of various ethnic groups living in the United States (from the Women's Health Study). *Am J Cardiol*. 2004;93(10):1238-1242. doi:10.1016/j.amjcard.2004.01.067
- Lundin JI, Peters U, Hu Y, et al. Methylation patterns associated with C-reactive protein in racially and ethnically diverse populations. *Epigenetics*. 2024;19(1):2333668. doi:10.1080/15592294.2024.2333668
- Reiner AP, Beleza S, Franceschini N, et al. Genome-wide association and population genetic analysis of C-reactive protein in African American and Hispanic American women. *Am J Hum Genet*. 2012;91(3):502-512. doi:10.1016/j.ajhg.2012.07.023
- Stowe RP, Peek MK, Cutchin MP, Goodwin JS. Plasma cytokine levels in a population-based study: relation to age and ethnicity. *J Gerontol A Biol Sci Med Sci*. 2010;65(4):429-433. doi:10.1093/gerona/glp198
- Vintimilla R, Hall J, Johnson L, O'Bryant S. The relationship of CRP and cognition in cognitively normal older Mexican Americans: A cross-sectional study of the HABLE cohort. *Medicine (Baltimore)*. 2019;98(19):e15605. doi:10.1097/MD.00000000000015605

- PRS: It is not clear why such a large proportion and effort is going towards the PRS calculation, which does not seem to be necessary in such detail and could easily be reduced to a standard PRS approach. Potential reasons for the PRS results (or lack of association) should also be elaborated on in the discussion.

Response: We strongly believe in using strong PRSs, and the choice of PRS method should not be affected by the analysis results (so that disappointing results should not mean that it suffices to use a "standard PRS approach"). Also, our lab has expertise in PRS, so for us it is not as difficult to develop them. Still, to address your comments, we moved some of the details of the PRS development to the supplement. In addition, we discussed the PRS results in the discussion section as follows:

“Our results validated that the PRS-CRP associates with blood CRP in a Hispanic/Latino population. On the other hand, the non-significant associations (accounting for multiple testing) observed for PRS-CRP with sleep and health outcomes indicate a weak relationship between genetic component of CRP with related comorbidities. Given that the PRS-CRP explained over 9% of the variance of circulating CRP levels, it is more likely that CRP is a consequence of these health conditions^{9,63}, rather than the other way around. That being said, there was a nominally significant association (OR=1.26, p-value=0.04) of the PRS-CRP constructed using a limited number of genome-wide significant SNPs, mimicking the one used by Huang et al.⁸, with moderate-to-severe OSA with EDS, compared with no OSA. Further research on the link between OSA and inflammation is needed to continue refining these results”.

- **hs CRP distribution:** Usually in the general population, the hs CRP values are skewed. How was determined if the log-transformation leads to a normally distributed variable and the assumptions for linear regression are not violated? The authors should also check the results presentation. A simple back transformation as indicated by the description of Figure 2 is not possible. It is rather more appropriate to report the effect as percent change or similar to make interpretation easier.

Response: Linear regression does not need to assume normally distributed errors. There is a result called “central limit theorem” that proves that under bounded moments of the outcome distribution, the effect estimates that come out of the linear regression are normally distributed and their standard errors are accurate. Of course, this is an asymptotic result. Still, our sample size is sufficient so that this central limit theorem holds. Please refer to Figure S1 for the density plot showing distribution of blood CRP before and after log-transformation.

You are completely correct about the effect size, we apologize for this oversight. We updated the wording as follows:

“MRS-CRP showed an almost two-fold higher association (effect size 0.36) than the two PRSs (effect size 0.2) with log-transformed blood CRP level. This means that each standard deviation (SD) increase in MRS-CRP corresponds to 43% percent increase blood CRP, while, one SD increase in PRS-wsum amounts to 23% percent increase in blood CRP level.”

- Adjusting for MRS: The rationale for this approach should be clarified. If the authors speculate that the CRP is the relevant mechanism (which is supported by the MRS but not PRS results) the approach to adjust for MRS might lead to collider bias, as adjustment on the causal chain might be problematic. Maybe other methods such as mediation analysis would be more appropriate.

Response: Thank you for the suggestion. We couldn't perform mediation analysis due to that sleep questionnaires, OSA assessments, blood sampling for CRP DNAm assays were all conducted

during the baseline visit. We did add extra analyses “Assessment of potential causal effect” (please refer to Methods and Results for details), where we performed 1) one-sample Mendelian Randomization and 2) association analysis incorporating MRS-CRP from visit 2 to explore potential causal and longitudinal associations. As a result, these analyses did not detect causal associations using our current data.

Nevertheless, the attenuated association between sleep outcomes and diabetes observed in our study after adjusting for MRS-CRP suggested that systemic inflammation may function as a common factor behind and possibly a consequence to these health conditions. We also believe that it’s of great benefits for future studies to study the directionality between DNA methylation and diseases onset.

- HCHS/SOL: The study design should be clarified. Are repeated measurements available? At which time-points (years) were the data collected?

Response: Thank you for the suggestion. We have added the time-points and statement regarding repeated measurements in the manuscript.

“The baseline HCHS/SOL exam took place during 2008-2011, and 16,415 participants were recruited. Obesity parameters BMI and waist-hip ratio were measured following standard procedure at baseline. Multiple sleep phenotypes were assessed using self-reported questionnaires from the baseline visit²⁶, with no repeated measurements.”

We don’t have repeated sleep measures nor repeated blood CRP measures, hence we only used data from the baseline exam. We do have repeated methylation measures, so we used MRS-CRP from the second visit to assess whether there is effect of diabetes and REI on increasing MRS-CRP. As explained in the updated manuscript:

Methods: “To assess whether there is effect of diabetes and REI on increasing MRS-CRP, we performed association analysis between the two phenotypes, modeled as exposure, and difference in MRS-CRP from Baseline to Visit 2, taking into account for model 1 covariates and time difference in years between the two visits.”

Results: “Neither diabetes (-0.12 [-0.24, 0.003]) nor REI (0.002 [-0.002, 0.006]) was found to be associated with changes in MRS-CRP between baseline and Visit 2. ”

- MRS calculation: There is a lot of effort in the PRS calculation. It is unclear why “only” one MRS was calculated and clumping and thresholding was not done here? The approaches used here should correspond for both scores. The participants from HCHS/SOL cohort study represented diverse Hispanic/Latino origins (and genetic admixture). However, the CpGs for constructing the MRS for CRP were derived from the reference 21 paper by Hillary et al., which **did not include the individuals of Latino race/ethnicity**. The potential impact of this should be addressed in the

discussion. It is also not clear why the authors then test again individual CpGs and focus on these results in the discussion. Both methods have their strength and limitations and the analysis should be focused on the approach which helps to address the research question.

Response: We calculated MRS based on pre-developed CpG sites from a recently published paper. And clumping and thresholding is the approach to select SNPs in PRS construction. Thank you very much for this great suggestion, we have updated the discussion accordingly:

“A previous study reported no association between average SpO2 and blood CRP in participants with OSA⁵³. Our findings in the general Hispanic population further support that average SpO2 is likely a less specific marker for OSA compared to MRS-CRP. Notably, since the CpG sites for MRS-CRP construction were developed using non-Hispanic participants²¹, our results supported not only the superior performance of MRS-CRP to circulating CRP in measuring chronic inflammation and related comorbidities^{18–21}, but also its transferability across different racial groups.”

Although the study conducted by Hillary et al. did not include participants from Hispanic background, our correlation results supported that their score outperforms others and validated its use in Hispanic population (Figure S3). Moreover, we searched through published studies regarding association of MRS-CRP with CRP. Though not many studies reported association between MRS-CRP with blood CRP level, our results are in alignment with a previous study (Stoldt et al. 2024) where one standard deviation increase in MRS-CRP was associated with a 0.38 unit increase in CRP in log scale.

In terms of discussing individual CpGs selected by Lasso regression, we want to provide some biological insights regarding mechanisms underlying these complex phenotypic traits.

Reference:

- Meike Stoldt, Farah Ammous, Lisha Lin, Scott M Ratliff, Erin B Ware, Jessica D Faul, Wei Zhao, Sharon L R Kardia, Jennifer A Smith, DNA Methylation at C-Reactive Protein-Associated CpG Sites May Mediate the Pathway Between Educational Attainment and Cognition, *The Journals of Gerontology: Series A*, Volume 79, Issue 8, August 2024, glae159, <https://doi.org/10.1093/gerona/glae159>

- Page 6, 2nd paragraph: please add the full name of PCs for the first time.

Response: Thank you for the suggestion. We have added the full name of PCs on page 6.

- Page 6, 3rd paragraph: “cohort study with participants of participants” participants was duplicated.

Response: Thank you for the correction. We have removed the redundant text on page 6.

- Sleep: why was a cutoff of 6 hours used to define short sleep, rather than the recommended 7 hours?

Response: We used 6 hours because it is typically used to define short sleep. More specifically, we also considered a few potential thresholds in prior work in the HCHS/SOL cohort (PMID 36493726) and concluded that a threshold of 6 hours is appropriate.

- Cell type proportions: this is very relevant. It should be stated in the methods how the cell type proportions were assessed.

Response: Thank you for the suggestion. We explained how cell type proportions were estimated in the methods section:

“Cell type proportions were estimated using a reference-based method⁴⁰ based on the genome-wide methylation data.”

- It is unclear which models were used. Is it always the “survey” approach? Why is this necessary? The weights are usually applied to obtain representative results, but this is not required here. Again, more details on the study design should be provided

Response: Yes, when using HCHS/SOL data we always applied the survey approach except for cubic spline model. We explained the reason in the methods section as follows:

“All association analyses were performed using survey linear regression models (survey package v4.2.1)⁴⁵ in R (v4.3.1)⁴² to account for the HCHS/SOL study design and generate inference applicable to the HCHS/SOL target population. Study design features addressed stratification, sampling weights, and non-response.

To account for potential U-shaped association of sleep duration with MRS- and blood CRP, we fitted penalized cubic spline models using mgcv package (v1.9.1)⁴⁷ in R (v4.3.1)⁴², adjusting for the same covariates as in model 1 using the primary analytic sample (2,221 individuals). The basis dimension for the spline term (k) was selected by minimizing the Akaike Information Criterion (AIC).”

Hope this helps clarify your question.

- “associations of PRS- and MRS-CRP with blood CRP level were first validated”: What validation has been done?

Response: We conducted association analysis using survey regression to confirm the significance of the associations between PRS- and MRS-CRP with blood CRP level. To avoid confusion, we updated the manuscript with following wording:

“We first validated the associations of PRS- and MRS-CRP with blood CRP level using 12,636 and 2,221 samples, respectively, adjusting for age, sex, BMI, study center, Hispanic/Latino background, and first 5 genetic PCs (model 1).”

- Confounder selection: The confounders for models 1 and 2 are reported twice on page 8. It is not clear why only a very basic set of confounders was chosen and other relevant factors such as smoking, physical activity, alcohol intake are not considered. Also, in addition to the lack of a PRS effect, the strong impact of waist-to-hip ratio indicates substantial residual confounding.

Response: Thank you for the suggestion. We have removed the redundant text on page 8. We have also added an additional sensitivity analysis adjusting for smoking, alcohol consumption and physical activity levels.

“Given that DNAm profile derived from whole blood represents an average across all present cell types, and the association of systemic inflammation with central obesity, smoking, alcohol and physical activities, we repeated association analyses for MRS-CRP adjusting for 1) cell-type proportions; 2) waist-hip ratio; 3) smoking and alcohol consumption status, as well as total time per day spent engaging in physical activities (measured in metabolic equivalent of task (MET)-minutes)”

- Role of ethnicity: It is highly appreciated that the present study is not limited to Caucasians only. As different PRS are calculated and tested, this aspect should also be acknowledged in the discussion

Response: Thank you! We have updated the discussion of the manuscript accordingly.

“This study demonstrates that omics measures of CRP perform well in association with CRP in a population of Hispanic/Latino adults, which were not examined before for these associations. Despite the heterogeneity (both genetic and lifestyle-related) of this population, there was no evidence of heterogeneity of biomarker association with CRP across self-reported Hispanic/Latino background groups.”

- Given the large number of analyses, the unit and effect estimate should always be given when reporting the results.

Response: Thank you for the comment. We have reported unit and effect size in the results section. For example,

“Interestingly, MRS-CRP is the only metric, modeled as exposure, to show statistically significant associations with REI (1.11 [0.36, 1.86]), minimum (-0.56 [-1.05, -0.08]) and average (-0.10 [-0.17,

-0.04]) SpO2 (%) (Figure 3A), implying that MRS-CRP correlates positively with OSA severity. Although not exhibiting significant associations with OSA traits, the coefficient of blood CRP level displays the same direction as that of MRS-CRP.

As shown in Figure 3B, MRS-CRP is also associated with higher risks of insomnia (odds ratio (OR) = 1.19 [1.02, 1.38]) and long sleep duration (OR = 1.29 [1.08, 1.55]), while blood CRP level is associated with lower risks of EDS (OR = 0.81 [0.68, 0.96]), longer overall sleep duration (0.13 [0.04, 0.21]), and increased risk of long sleep duration (OR = 1.31, [1.13, 1.53]). Meanwhile, short sleep duration is not significantly associated with any of the CRP measures (Figure 3B). On the other hand, PRS-CRPs are not associated with any sleep traits (Figure 3)."

Hope this helps clarify.

- Multiple testing: Why did the authors use FDR adjustment? How was this calculated with the large number of different models? This is usually applied for explorative analysis. In this case, a clear hypothesis is tested. The Bonferroni correction with the number of independent tests might be more appropriate.

Response: Thank you for the comment. We chose the FDR adjustment because it is more powerful and less restricted than Bonferroni when it comes to large numbers of comparisons. This allows us to detect more true effects. Importantly, the tests are not all independent, because different CRP measures are correlated with each other, and some sleep measures are correlated with each other – making Bonferroni correction a very conservative alternative, potentially resulting many false negatives. We now explain this in the methods:

"The FDR correction was employed to account for multiple testing, as the correlations among CRP measures and among sleep measures lead to non-independent tests, which could result in overly conservative Family-Wise Error Rate adjustment."

Note that FDR correction also tends to be conservative when there are correlated tests. However, it is still less conservative than FWER correction such as Bonferroni procedure.

- Page 9, last paragraph: "Overall, women constitute 65.8% of this sample, with higher female percentage, 75.7%, among participants with blood CRP level larger than 3 mg/L." The numbers in this sentence were not consistent with those in the Table 1. For example, women constitute 65.8% of this sample, but the Table 1 shows 1435 females, and the percentage should be $1435/2221 = 64.6\%$. This percentage 1435 (58.4%) in Table 1 was also not correct. "75.7%, among participants with blood CRP level larger than 3 mg/L" was also wrong. Many numbers and percentages in Table 1 do not match and should be double-checked.

Response: Thank you! We didn't realize that the percentage numbers are misplaced. We have updated the table with correct percentages.

- Page 13, Figure 3: The text on the right is not aligned with the forest plot on the left. Full names of the abbreviations should be added in the footnote.

Response: Thank you for the suggestion. We have updated the Figure 3 with added abbreviations on the footnote.

- Page 14, 3rd and 4th paragraphs: several papers were cited to compare the findings with previous research. Since this is the Results section, it would be more appropriate to focus solely on describing the results rather than including discussions or comparisons with other studies.

Response: Thank you for the suggestion. We have rewritten this part and moved the reference to discussion. Below is the updated text:

“Our cubic spline model results showed a J-shaped association between sleep duration and CRP markers where optimal sleep time (around 7 hours) corresponds to lowest level of both MRS- and blood CRP level (Figure S7). As sleep duration extends, the increase in levels of CRP markers become more pronounced (Figure S7).”

- Figure S5: The figure shows a J shaped association, although this does not seem to be significantly different from a linear association. What is the estimated df? Was the non-linear term significant?

Response: Yes, the non-linear term is significant. The model degree of freedom is 5.

Discussion

- The discussion should start with an overall summary of the findings to address the hypothesis and include a strength and limitations section.

Response: Thank you for the advice, we have added a summary section at the beginning of the discussion.

“This study demonstrates that omics measures of CRP perform well in association with CRP in a population of Hispanic/Latino adults, which were not examined before for these associations. Despite the heterogeneity (both genetic and lifestyle-related) of this population, there was no evidence of heterogeneity of biomarker association with CRP across self-reported Hispanic/Latino background groups. We validated the stronger associations of MRS-CRP with long-term sleep and metabolic conditions, such as OSA and diabetes, compared to blood CRP, which showed modest associations. Adjusting for MRS-CRP attenuated the associations between sleep traits and metabolic conditions, suggesting systemic inflammation as either a mediating or a shared causal factor. However, in contrast to MRS-CRP, PRS-CRP exhibited minimal associations with sleep and related health outcomes, suggesting limited role in influencing these inflammation-related phenotypes. Further, one-sample Mendelian randomization analysis did not support a causal association of CRP on neither sleep, diabetes, nor hypertension. Overall, this work demonstrates strong association of inflammation, modeled via MRS-CRP, with sleep-related health outcomes. Secondary analysis selecting specific CpG methylation sites in association with sleep-related and metabolic outcomes also advances the understanding of potential epigenetic mechanisms underlying the relationship between these health outcomes and inflammation.”

As for strength and limitations, we have spread them out throughout the discussion. For example, we discussed the limitations of our study: “Future work should assess potential mediation using longitudinal data. Another limitation of current study stems from the potential bias in self-reported sleep measurements, which often fall short of capturing context-specific sleep patterns influenced by environmental and lifestyle factors.”

We also talked about the significance and strengths: “Our results validated that the PRS-CRP associates with blood CRP in a Hispanic/Latino population”. “We validated the stronger associations of MRS-CRP with long-term sleep and metabolic conditions, such as OSA and diabetes, compared to blood CRP, which showed modest associations.”. “Notably, since the CpG sites for MRS-CRP construction were developed using non-Hispanic participants²¹, our results supported not only the superior performance of MRS-CRP to circulating CRP in measuring chronic inflammation and related comorbidities¹⁸⁻²¹, but also its transferability across different racial groups.”

- The direction and strength of the effect and causality should be discussed.

Response: Thank you for the suggestion. The direction and strength of the effect is mentioned in the results. For example:

“Interestingly, MRS-CRP is the only metric, modeled as exposure, to show statistically significant associations with REI (1.11 [0.36, 1.86]), minimum (-0.56 [-1.05, -0.08]) and average (-0.10 [-0.17, -0.04]) SpO2 (%) (Figure 3A), implying that MRS-CRP correlates positively with OSA severity. Although not exhibiting significant associations with OSA traits, the coefficient of blood CRP level displays the same direction as that of MRS-CRP.

As shown in Figure 3B, MRS-CRP is also associated with higher risks of insomnia (odds ratio (OR) = 1.19 [1.02, 1.38]) and long sleep duration (OR = 1.29 [1.08, 1.55]), while blood CRP level is associated with lower risks of EDS (OR = 0.81 [0.68, 0.96]), longer overall sleep duration (0.13 [0.04, 0.21]), and increased risk of long sleep duration (OR = 1.31, [1.13, 1.53]). Meanwhile, short sleep duration is not significantly associated with any of the CRP measures (Figure 3B). On the other hand, PRS-CRPs are not associated with any sleep traits (Figure 3).”

We added extra analysis “Assessment of potential causal effect”, where we performed 1) one-sample Mendelian Randomization and 2) association analysis incorporating MRS-CRP from visit 2 to explore potential causal and longitudinal associations (please refer to Methods and Results for details).

Hope this helps clarify.

- What is the novel aspect and clinical relevance? CRP is already included in the standard clinical routine, what could be improved here or would this provide a therapeutic target?

Response: Our results showed a stronger association for DNA methylation risk score of CRP (MRS-CRP) with certain sleep and metabolic outcomes than that of CRP itself, hence supporting that incorporation of MRS-CRP could potentially improve population risk stratification for these adverse health outcomes. Through analyzing specific CpGs selected by Lasso regression, our findings also provided insights in biological mechanisms underlying the connection of sleep,

metabolic syndrome and inflammation. Furthermore, we believe that these CpGs could be potentially used as biomarkers to further derive therapeutic targets.

As described in the manuscript: “Overall, these lasso-selected CpG sites reiterate the important role of DNAm in the pathogenesis underlying complex diseases, and may serve as potential biomarkers for inflammatory stress associated with metabolic syndrome and OSA traits, bridged by fat, glucose and energy homeostasis.”

“A CRP measure based on DNA methylation status at select CpG sites shows stronger relationship with circulating CRP level than PRS-CRP. Moreover, MRS-CRP outperforms both genetic and circulating CRP in association with several OSA-related sleep traits and metabolic comorbidities, while its associations with other sleep and cognitive measures were mixed. Our results also suggest a possible shared mechanism between sleep phenotypes and metabolic syndrome, related to systemic inflammation and modeled by MRS-CRP, particularly diabetes. Consequently, the incorporation of MRS-CRP, versus circulating CRP, in clinical assessment has the potential to improve population risk stratification for adverse health outcomes.”

- “PFDN2-cg00816397, ATF2-cg02622866 and ARSA-cg07453718” were not found in the Table 2.

Response: These CpGs were listed in Table S7-S8, as Table 2 reported the “CpGs selected by Lasso regression across obstructive sleep apnea and metabolic comorbidities”. As specified in the discussion section, these three CpG sites are exclusive for OSA.

“The large effect sizes of some CpGs for REI and minimum SpO₂, such as MKNK2-cg03246954, PFDN2-cg00816397, ATF2-cg02622866 and ARSA-cg07453718, indicate strong associations with OSA, and potential applicability as disease biomarkers (Table S8-S9).”

- The conclusion should be revised and focused on the results

Response: Thank you for the suggestion, we have updated the conclusion:

“A CRP measure based on DNA methylation status at select CpG sites shows stronger relationship with circulating CRP level than PRS-CRP. Moreover, MRS-CRP outperforms both genetic and circulating CRP in association with several OSA-related sleep traits and metabolic comorbidities, while its associations with other sleep and cognitive measures were mixed. Our results also suggest a possible shared mechanism between sleep phenotypes and metabolic syndrome, related to systemic inflammation and modeled by MRS-CRP, particularly diabetes. Consequently, the incorporation of MRS-CRP, versus circulating CRP, in clinical assessment has the potential to improve population risk stratification for adverse health outcomes.”

Reference

References are in various formats, which should be harmonised.

Response: Thank you for the comment. We have harmonized the reference format.

Reviewer #2 (Remarks to the Author):

Peer-review for: Analysis of C-reactive protein omics-measures associates methylation risk score with sleep health and related health outcomes

This is an interesting study concept on how sleep apnea is related to chronic inflammation as measured by CRP. My main concerns are twofold. First, I'm not sure that the MRS for CRP was constructed adequately. Second, my interpretation of the plots is slightly different than those of the authors (I see mostly null results). I attempted to also one avenue to improve some of the expected biases, which I believe would be easy to perform by the authors.

Major points:

1) It is well explained that the CRP PRS was built in a separate cohort (MESA) than the one it was tested on (HCHS/SOLA). The authors actually take great care in ensuring that some their analyses do not overfit. However, my current understanding of the MRS is that it was computed on the HCHS/SOL cohort, the same one it was tested on. Is that so? This is important, as if it were the case then the association between MRS and the different traits could be falsely amplified.

Response: Thank you for the comment. The weights for MRS construction were obtained from a previous study, so we are not worried about the overfit problem in this case. We updated our wording to better explain the MRS-CRP calculation in Methods section:

“Methylation risk scores (MRS) of CRP were constructed as weighted sums of each individual's DNAm values of pre-developed CpG sites²¹ using MethParquet package (v0.1.0)⁴¹ in R (v4.3.1)⁴². To validate that the most recently reported MRS-CRP (Hillary et al.²¹) is also the most powerful MRS-CRP in Hispanic/Latino adults, we constructed multiple MRS-CRPs that were previously reported^{18,21,43,44}, and computed the correlation between blood CRP and the MRSs. For association analyses, the constructed MRS-CRP was standardized across the sample by subtracting the mean before dividing it by its standard deviation.”

2) When I look at Figure 3, I feel like the number of bars that do not cross the null is grossly expected for the number of analyses. Further, even if assuming that no multiple testing errors are shown, I have a hard time wrapping my head around some results. For example, if the CRP PRS is indeed associated with higher AHI, then how to reconcile this with a lower daytime sleepiness?

Response: We performed multiple testing correction, and therefore we are confident in the results. We would like to clarify that one can think of testing in the lens of whether a hypothesis was rejected or not, but it is also important to consider the strength of association, and both are accounted for in multiple testing procedures. In sum, we believe that the observed associations and their statistical significance are appropriate. If anything, we are likely being conservative because many of the tests are correlated and multiple testing procedures tend to be conservative

in the presence of correlation. There are approaches to account for it, but they do not readily apply for the settings of multiple outcomes and multiple exposures, therefore, we applied multiple testing correction conservatively.

With respect to the PRS: it was only associated with long sleep. It was not associated with REI (now changed to be named REI), nor with excessive daytime sleepiness. Blood CRP had protective association with EDS, and a non-significant association with higher REI. While we expect the latter, the protective association with EDS is indeed surprising. Overall, this perhaps underscores the weaker, and perhaps more “noisy” association of blood CRP with sleep phenotypes.

3) One way to potentially alleviate some of the problems above would be to use the CRP PRS as a statistical instrument in a one-sample MR. That is measure the PRS calculated in MESA in the HCHS/SOL participants. Then use a two-stage linear regression model with this calculated PRS. Since the authors have primary data access, this should be easy to do using the appropriate statistical package (R has good two-stage linear regression packages, for instance). This is a rigorous way of doing one-sample Mendelian randomization, assuming that the PRS explains enough of the variance in CRP (i.e. has an F-value greater than 10, as a rule of thumb).

Response: Thank you for this great suggestion! We have performed one-sample MR for both MRS-CRP and blood CRP with PRS-CRP as the statistical instrument and reported the results in Table S12. The F-statistics of the two PRS-CRPs for MRS-CRP, however, were below 10, indicating that the causal estimate for MRS-CRP would not be reliable with PRS-CRP as a statistical instrument. And our results showed no statistical significance for causal associations between blood CRP and the sleep traits.

Reviewer #3 (Remarks to the Author):

This is a well-executed and technically sound study. The authors have applied appropriate and timely analytical approaches throughout. While they did not find an association between their genetic risk score and health-related outcomes, this result is intriguing in itself. Furthermore, their identification of novel associations between the methylation risk score and sleep-related outcomes is particularly noteworthy, as well as the expected associations with cardiometabolic outcomes. Overall, the study offers valuable insights and makes a meaningful contribution to the field.

As I understand it, there are three major CRP EWAS meta-analyses. It would be valuable to assess the stability of the MRS-CRP score based on 1,041 CpGs from Hillary et al. (2024) in comparison to the 218 CpGs identified by Ligthart et al. (2016) and the 1,511 CpGs reported by Wielscher et al. (2022). Specifically, it would be informative to examine how these scores, derived from earlier EWAS studies, are associated with both blood CRP levels and sleep traits within the HCHS/SOL study.

Response: Thank you for the suggestion. To prevent overfitting and reduce multiple testing burden, we now report results from association analysis of these other MRS-CRP scores with blood CRP (and not with sleep phenotypes).

We didn't perform the benchmarking selection for CpGs as the 1,511 CpGs reported by Wielscher et al. (2022) were already compared in the work of Hillary et al. (2024), while the 218 CpGs reported by Ligthart et al. (2016) were assessed for replication rate and effect size in the work of Wielscher et al. (2022). However, we did compare their correlations with blood CRP within the HCHS/SOL study, which is now reported in our manuscript as Figure S3 to better clarify their performance in this population. Thanks again for the suggestion!

It would also be highly interesting to explore the differences in CRP associations between the PRS and MRS across different ethnicities. Stratifying the MRS ~ blood CRP associations by the detailed population background in your study could provide valuable insights and would be a compelling addition to the analysis.

Response: Thank you for the comment. It would indeed be intriguing to see the results by detailed population background. However, given the current sample size (about 2000 people in total), we worry that for some backgrounds the number of participants would be too small to repeat the analysis. It would be great to do so in the future with more data becoming available. That said, we conducted Cochran's heterogeneity test for associations of MRS-CRP, PRS-CRP with blood CRP and found no heterogeneity across distinct ethnic backgrounds. This result is reported in the manuscript under the results section "CRP omics-markers validation":

“Cochran’s heterogeneity test suggests homogeneity of both MRS-CRP (p-value = 0.33) and PRS-CRP (p-value = 0.8) across Hispanic/Latino backgrounds in association with blood-CRP (see Table S6 for group-specific effect estimates).”

For Figure 3, please add clarifications in the legend to define terms such as AHI and mean SpO₂, ensuring that the figure can be understood independently. Additionally, since the PRS is not significantly associated with any outcomes, I suggest removing the PRS weighted sum associations, as they do not seem to contribute meaningfully to the narrative. Lastly, the alignment of the forest plot with the table on the left side is off, making it difficult to read; this should be adjusted for better clarity.

Response: Thank you for the suggestion. We have added the full names for AHI (now renamed to REI) and SpO₂ measures in the figure and table legends. Also, although the PRS is not significantly associated with some health outcomes, it is significantly associated with circulating CRP level, which means that the PRS developed using primarily European population is transferable to Hispanic population. So, we believe it is still worth reporting as part of the results.

For Figure 1, it would be helpful to clarify the source of the input CpGs for the MRS—specifically, indicating which paper they are derived from. Additionally, the figure would benefit from a clearer distinction between the processes involving the PRS and MRS, as well as the datasets used for each. This would enhance the figure’s overall clarity and make it easier for readers to follow the methodology.

Response: Thank you very much for the suggestion. In this revision, we created a new Figure 1, and revised the manuscript such that the development of the PRS is described mostly in the supplement (and does not appear in Figure 1), and Figure 1 focuses on association analyses rather than on development and construction of CRP measures. The source of CpG weights for MRS was referenced in the methods section:

“Methylation risk scores (MRS) of CRP were constructed as weighted sums of each individual’s DNAm values of pre-developed CpG sites²¹ using MethParquet package (v0.1.0)⁴¹ in R (v4.3.1)⁴².”

Since you have already conducted an association analysis between sleep outcomes and diabetes, adjusting for MRS, it would be valuable to also perform a mediation analysis. This could help determine whether the mediation effect of the MRS on the relationship between sleep outcomes and diabetes is significant.

Response: Thank you very much for the suggestion. We completely agree in principle, however, as much as we would like to perform a mediation analysis, our data is limited to association analysis, due to that sleep questionnaires, OSA assessments, blood sampling for CRP DNAm assays were all conducted during the baseline visit. Nevertheless, we added extra analysis “Assessment of potential causal effect”, where we performed 1) one-sample Mendelian Randomization and 2) association analysis incorporating MRS-CRP from visit 2 to explore potential causal and longitudinal associations (please refer to Methods and Results for details).